# On Optimal Caching and Model Multiplexing for Large Model Inference

**Banghua Zhu**
Department of EECS
UC Berkeley
banghua@berkeley.edu

**Ying Sheng**
Computer Science Department
Stanford University
Ying.Sheng@stanford.edu

**Lianmin Zheng**
Department of EECS
UC Berkeley
lmzheng@berkeley.edu

**Clark Barrett**
Computer Science Department
Stanford University
barrett@cs.stanford.edu

**Michael I. Jordan**
Department of EECS
UC Berkeley
jordan@berkeley.edu

**Jiantao Jiao**
Department of EECS
UC Berkeley
jiantao@berkeley.edu

## Abstract

Large Language Models (LLMs) and other large foundation models have achieved noteworthy success, but their size exacerbates existing resource consumption and latency challenges. In particular, the large-scale deployment of these models is hindered by the significant resource requirements during inference. In this paper, we study two approaches for mitigating these challenges: employing a cache to store previous queries and learning a model multiplexer to choose from an ensemble of models for query processing.

Theoretically, we provide an optimal algorithm for jointly optimizing both approaches to reduce the inference cost in both offline and online tabular settings. By combining a caching algorithm, namely Greedy Dual Size with Frequency (GDSF) or Least Expected Cost (LEC), with a model multiplexer, we achieve optimal rates in both offline and online settings. Empirically, simulations show that the combination of our caching and model multiplexing algorithms greatly improves over the baselines, with up to $50\times$ improvement over the baseline when the ratio between the maximum cost and minimum cost is 100. Experiments on real datasets show a $4.3\times$ improvement in FLOPs over the baseline when the ratio for FLOPs is 10, and a $1.8\times$ improvement in latency when the ratio for average latency is $1.85$.

## 1 Introduction

The recent emergence of Large Language Models (LLMs) and foundation models has significantly increased the capabilities of AI systems (Bubeck et al., 2023; Nori et al., 2023; Ziegler et al., 2019; Ouyang et al., 2022; OpenAI, 2023; Beeching et al., 2023; Chowdhery et al., 2022; Wei et al., 2022a; Google, 2023). This progress comes at a cost, however, of increased resource consumption and latency during both training and inference, presenting challenges not only in real-world deployment but also in terms of environmental impact and energy usage (Sharir et al., 2020; Patterson et al., 2021; Bommasani et al., 2022). For instance, LLM-based chatbots typically consist of large transformer-based networks with parameter counts ranging from one to several hundred billion (Zhou et al., 2023). Moreover, the auto-regressive nature of LLMs exacerbates the issue of latency and resource consumption because the model can only generate one token at a time. Thus, compared to traditional AI-powered services, language model inference costs are much higher and the latency is significantly

37th Conference on Neural Information Processing Systems (NeurIPS 2023).

longer, making it nearly impossible to process each query using LLMs in high-throughput query systems such as search engines.

In this paper, we explore two simple yet effective strategies to mitigate this problem: (1) employing a caching system to store previous queries, and (2) developing a model multiplexer to choose the most appropriate model from a set of models for processing the queries. The general workflow of our proposed LLM-based inference system is shown in Figure 1: upon receiving a query or prompt, we initially check if it can be retrieved from the cache. If the query is not found in the cache, we employ the model multiplexer to determine which model should be used for processing it first, based on the estimated cost for both models.

The choice of cost function and models can vary based on the goal. One measure of cost, for example, could be floating point operations (FLOPs). Other alternatives could include the number of API calls as a measure of resource consumption, latency as a measure of time consumption, or a score provided by a user as a measure of user satisfaction. The cost could also be a weighted sum of multiple factors. For the models, a natural choice would be to have a small and a large model, where the small model costs less and is also less accurate, and the large model has a higher cost and also provides higher accuracy. Another alternative would be to have models with expertise in different areas, i.e., each model has high accuracy in its own area of expertise. We provide more discussion in Appendix A.

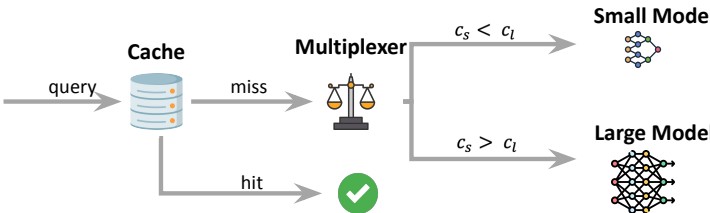

Figure 1: A workflow for LLM-based inference with caching and model multiplexing.

There is a long history of existing literature on caching algorithms, with prominent applications including computer architecture and web retrieval (Smith, 1982; Wang, 1999; Kumar and Singh, 2016). Existing caching algorithms deal with queries with different frequencies and cost, and must also provide guidelines for choosing the cache size. In addition to these well-known difficulties, the use of caching for LLMs raises new challenges, including:

- **The need for fuzzy search.** Since the prompt lies in a discrete space that is exponentially large with respect to the token size, it is impossible to match and save all distinct queries. Thus, to be at all useful, approximate matching and grouping is required when retrieving queries saved in the cache.

- **The randomness of the cost.** The cost for processing each query is a random variable that depends on the query and has a large variance due to the auto-regressive generation procedure and the difference in the length and quality of generated responses. When combined with the long-tailed distribution of the query frequency, the estimation of the cost requires a non-trivial algorithm design.

- **The effect of model multiplexing.** When the cache system is combined with the model multiplexer, the estimation of cost must change accordingly to take into consideration the different costs induced by various models.

For the fuzzy search problem, semantic search or vector-embedding-based ideas provide a systematic solution that includes embedding extraction and matching algorithms (Bast et al., 2016; Chang et al., 2020; Kamalloo et al., 2023). To simplify the problem, we assume that there exists some semantic search oracle that can group the prompts with the same semantic meaning and that the total cache size is limited by the number of queries, ignoring the difference in cache size between each individual query and response.

The remainder of this paper is organized as follows. In Section 2, we formally define the pipeline of caching and model multiplexing. In Section 3, we study the optimality of the Least Expected Cost (LEC) caching strategy, which estimates the frequency and cost of processing each query, and evicts the one with the least estimated expected cost when there is only one model to call. In section 4, we

consider the case when we have access to two models, and jointly design optimal caching and model multiplexer. In both sections, we start by assuming there are infinite samples and then analyze the offline and online learning cases where the cost and frequency need to be learned from data. The experimental results are presented in Section 5. We discuss the potential choices of cost, model, and output in the real world in Appendix A. We provide a brief discussion of the generalization to variable cache sizes in Appendix B and of the generalization to multi-model multiplexing in Appendix C.

## 1.1 Related work

**Cache replacement algorithms** Traditional cache replacement algorithms investigate optimal ways to cache queries with different frequencies, costs, and cache sizes. To address varying frequencies, a standard approach is to use a Least Frequently Used (LFU) or Least Recently Used (LRU) cache eviction strategy (Lee et al., 2001). These have been proven to be optimal for both adversarial and stochastic queries (Stallings and Paul, 2012; Bura et al., 2021). Caching has also been combined with machine learning advice and online learning analysis in the literature (Chang et al., 2018; Shuja et al., 2021; Jiang et al., 2019; He et al., 2017; Mukhopadhyay and Sinha, 2021; Faizal et al., 2023). When varying costs and varying frequencies exist simultaneously, Jin and Bestavros (2000); Arlitt et al. (2000) propose and study the Greedy Dual-Size with Frequency (GDSF) replacement algorithm, which takes both frequency and cost into consideration. Bahn (2005) proposes the Least Expected Cost (LEC) algorithm, which is similar to GDSF, except that it estimates frequency from data. Our work extends this idea by attempting to learn a model for both frequency and cost from data. Moreover we explore the statistical optimality of these algorithms in both offline and online settings. We also investigate combining caching algorithms with model multiplexing in order to boost performance.

**Acceleration of LLM inference** Much effort has been devoted to reducing the cost and latency of LLMs during inference. For example, post-training quantization-based approaches aim to compress the model size by using lower-precision arithmetic without losing too much accuracy (Gholami et al., 2021; Frantar et al., 2023). Early-exit frameworks aim to utilize the output in the middle decoder blocks so that only a small fraction of decoder blocks are called when processing a query (Bakhtiarnia et al., 2022; Schuster et al., 2022). The Mixture of Experts approach designs a gating function that only assigns a small fraction of the network for each query (Fedus et al., 2022). Embedding recycling caches activations from an intermediate layer of a pre-trained model to accelerate the training and inference procedure (Du et al., 2020; Wei et al., 2022b; Saad-Falcon et al., 2023). LLM cascade starts with the smallest model and continues to call larger models if the output is not acceptable (Chen et al., 2023b). The big little transformer decoder framework uses a smaller model to generate a draft response and calls the large model to identify the unreliable tokens and perform correction (Kim et al., 2023). Similar ideas have been combined with speculative sampling to guarantee that the output remains the same in distribution as that of the large models (Chen et al., 2023a; Leviathan et al., 2022).

## 2 Formulation

We formalize the workflow in Figure 1. Consider the set of (finite) prompts / queries $\mathcal{Q} \subset \mathbb{R}^d$. In the $t$-th round, a query $q_t \in \mathcal{Q}$ is sampled from a fixed population distribution $P \in \Delta(\mathcal{Q})$. We maintain a small set of cache $\mathcal{L}_t \subset \mathcal{Q}$ with $|\mathcal{L}_t| \leq L$. We say the query hits the cache if the query satisfies $q_t \in \mathcal{L}_t$. When the query hits the cache, the incurred cost is zero. When the query does not hit the cache, we choose among the existing models to process the query.

In the processing stage, we first describe the setting of caching without model multiplexing, and extend it to the case of caching with model multiplexing.

### 2.1 Caching without model multiplexing

In the case when we only have one model, let $C_l(q)$ denote the random variable of the cost when processing the query with the model. Assume that $C_l(q)$ is supported on $[B_1, B_2]$ with $B_2 > B_1 > 0$ being the upper and lower bounds for the cost. Let $c_l^\star(q) = \mathbb{E}[C_l(q)]$ be the expected true cost of

processing the query $q$. The cost for a given query $q$ and cache $\mathcal{L}$ can be written as:

$$\text{cost}(q, \mathcal{L}) = \mathbb{1}(q \notin \mathcal{L})\mathbb{E}[C_l(q)] = \mathbb{1}(q \notin \mathcal{L})c_l^\star(q).$$

By taking the expectation over the distribution $q$, we have the expected cost as

$$\text{cost}(\mathcal{L}) = \sum_q P(q)\mathbb{1}(q \notin \mathcal{L})c_l^\star(q).$$

In the offline learning setting, we collect an offline dataset and hope to learn a caching policy $\widehat{\mathcal{L}}$ such that $\text{cost}(\widehat{\mathcal{L}})$ is minimized.

In the online setting, the query comes in a streaming fashion. At the beginning of each round, we receive a query $q_t$. If the query misses the current cache $\mathcal{L}_t$, we let the model process the query and receive a cost $c_t \sim \mathbb{P}_{C_l}$. Then we can choose to update the cache $\mathcal{L}_t$ by adding the current query and response to the cache, and replacing one of the existing cached items if the cache $\mathcal{L}_t$ is full. If the query hits the cache $q_t \in \mathcal{L}_t$, then the cost for this round is set to zero with no more observations. In this case, we are interested in characterizing the average difference in the cost throughout the execution of the online learning process. This can be characterized by the regret:

$$\text{Regret}_{\text{cache}}(T) = \sum_{t=1}^{T} \mathbb{E}[\text{cost}(q_t, \mathcal{L}_t) - \text{cost}(q_t, \mathcal{L}^\star)].$$

## 2.2 Caching with model multiplexing

For the simplicity of the notation, we focus on the case of selecting from a small model and a large model,[1] and discuss how it can be generalized to the case of selecting from multiple models in Appendix C. Let $C_s(q)$ denote the random variable of the cost when processing the query with the small model, and $C_l(q)$ denote the random variable of the cost when processing the query with the large model. We assume that both random variables are supported on $[B_1, B_2]$. We observe $i.i.d.$ draws of the random variables $C_s(q)$ when executing the small model, and $C_l(q)$ when executing the large model. Denote the expected cost as $c_s^\star(q) = \mathbb{E}[C_s(q)]$ and $c_l^\star(q) = \mathbb{E}[C_l(q)]$.

Let $\pi : \mathcal{Q} \mapsto [0, 1]$ be the (possibly random) model multiplexing policy that maps the query $q$ to values in $[0, 1]$, where $\pi(q) = 1$ represents that the query is always sent to the small model, and $\pi(q) = 0$ represents the query is always sent to the large model. The randomness in the policy $\pi$ is independent of the cost $C_s(q), C_l(q)$. The total cost can be written as the following function of the query $q$, cache $\mathcal{L}$ and policy $\pi$:

$$\begin{aligned}
\text{cost}(q, \mathcal{L}, \pi) &= \mathbb{1}(q \notin \mathcal{L})\mathbb{E}[C_s(q)\pi(q) + C_l(q)(1 - \pi(q))] \\
&= \mathbb{1}(q \notin \mathcal{L})(c_s^\star(q)\pi(q) + c_l^\star(q)(1 - \pi(q))).
\end{aligned}$$

By taking the expectation over $q$, we have the expected cost as

$$\text{cost}(\mathcal{L}, \pi) = \sum_q P(q)\mathbb{1}(q \notin \mathcal{L})(c_s^\star(q)\pi(q) + c_l^\star(q)(1 - \pi(q))).$$

In the offline learning setting, we collect an offline dataset and hope to learn a caching policy $\widehat{\mathcal{L}}$ and a multiplexer $\hat{\pi}$ such that $\text{cost}(\widehat{\mathcal{L}}, \hat{\pi})$ is minimized. In the online setting, we get to update the cache in each round by adding the current query into the cache and evicting the ones in the cache if full. When the query $q_t$ misses the cache in round $t$, we will observe a sample from $C_s(q_t)$ if it is processed by the small model, or a sample from $C_l(q_t)$ if it is processed by the large model. There will be no observations of cost if $q_t$ hits the cache. We aim at minimizing the regret:

$$\text{Regret}_{\text{sel}}(T) = \sum_{t=1}^{T} \mathbb{E}[\text{cost}(q_t, \mathcal{L}_t, \pi_t) - \text{cost}(q_t, \mathcal{L}^\star, \pi^\star)].$$

---

[1] Note that although we name the models as small and large models, we do not impose any assumption on the relationship between their costs. Moreover, the model size and cost function can be arbitrary for both models.

# 3 Optimal Caching without Model multiplexing

## 3.1 Population setting

We start with the population setting where the probability distribution $P$ and the cost $c_l^\star$ are both known. In the case with only one model, the optimal caching strategy is the Least Expected Cost (LEC) or Greedy Dual Size with Frequency (GDSF) algorithm:

$$\mathcal{L}^\star = \mathcal{L}_{\mathsf{LEC}} = \arg\min_{\mathcal{L}:|\mathcal{L}|\le L} \mathsf{cost}(\mathcal{L}) = \arg\min_{\mathcal{L}:|\mathcal{L}|\le L} \sum_{q\in\mathcal{Q}} P(q)\mathbb{1}(q\notin\mathcal{L})c_l^\star(q).$$

The traditional frequency-based caching strategy, including Least Recent Used (LRU) and Least Frequently Used (LFU), aims at caching the most frequent queries:

$$\mathcal{L}_{\mathsf{LFU}} = \arg\min_{\mathcal{L}:|\mathcal{L}|\le L} \sum_{q\in\mathcal{Q}} P(q)\mathbb{1}(q\notin\mathcal{L}).$$

We show in Appendix D that the ratio between the cost of LFU and LEC can be as high as $\frac{\max_{q\in\mathcal{Q}} c_l^\star(q)}{\min_{q\in\mathcal{Q}} c_l^\star(q)}$ in the worst case, which shows that LFU can be highly suboptimal when the cost varies significantly.

## 3.2 Finite sample setting: Offline learning

The previous section characterizes the optimal caching strategy in the population setting. We now consider the finite-sample offline learning setting, where we hope to produce a cache $\mathcal{L}$ based on prior data such that the introduced cost is minimized. Denote $\mathcal{D}_N = \{(q_1, c_1), \cdots, (q_N, c_N)\}$, where $q_i$ is sampled from the distribution $P(\cdot)$, and $c_i$ is a sample from random variable $C_l(q_i)$. We consider estimating $P, c_l^\star$ from oracles $\hat{P} = \mathsf{DenEstOracle}(q_1, \cdots, q_N)$, $\hat{c}_l(q) = \mathsf{RegressionOracle}(\mathcal{D}_N)$. In practice, one may remove the last layer of the pre-trained language model and concatenate it with a linear head and fine-tune the model as the estimator. For theoretical analysis, we focus on the tabular case, where we set both $\hat{P}$ and $\hat{c}_l(q)$ to be the plug-in estimator:

$$\hat{P}(q) = \frac{\sum_{i=1}^N \mathbb{1}(q_i = q)}{N}, \tag{1}$$

$$\hat{c}_l(q) = \begin{cases} \frac{\sum_{i=1}^N \mathbb{1}(q_i=q)c_i}{\sum_{i=1}^N \mathbb{1}(q_i=q)}, & \text{if } \sum_{i=1}^N \mathbb{1}(q_i=q) > 0 \\ B_1, & \text{if } \sum_{i=1}^N \mathbb{1}(q_i=q) = 0. \end{cases} \tag{2}$$

In practice, the distribution of $q$ may have a long tail. Although the estimation of $P(q)$ is uniformly good for all $q$, the estimation of $c^\star(q)$ can be bad for the queries that are visited less. To select the maximum $L$ elements from the imbalanced samples, we compensate the plug-in estimator by introducing pessimism (Rashidinejad et al., 2021; Jin et al., 2021)[2]. As we show in Lemma 1, the true frequency for any query $q \in \mathcal{L}^\star$ is lower bounded by some constant that depends on $B_1, B_2, |\mathcal{Q}|$. Thus the pessimism helps eliminate those less visited queries in the long tail of the distribution and encourages caching the queries in $\mathcal{L}^\star$. The lower-confidence-bound based estimator is:

$$\hat{\mathcal{L}} = \arg\min_{\mathcal{L}:|\mathcal{L}|\le L} \sum_{q\in\mathcal{Q}} \mathbb{1}(q\notin\mathcal{L})\hat{P}(q) \cdot \max\left(B_1, \left(\hat{c}_l(q) - (B_2 - B_1)\sqrt{\frac{\log(6N|\mathcal{Q}|/\delta)}{2\sum_{n=1}^N \mathbb{1}(q_n = q)}}\right)\right).$$

We show how the cost for the caching from the empirical estimate differs from the optimal cost.

**Theorem 1.** *Assume that $N \ge \frac{8B_2|\mathcal{Q}|\log(3L/\delta)}{B_1}$ and taking $\delta = 1/N$. We have*

$$\mathbb{E}[\mathsf{cost}(\hat{\mathcal{L}}) - \mathsf{cost}(\mathcal{L}^\star)] \le C(B_2 - B_1)L \cdot \sqrt{\frac{B_2|\mathcal{Q}|\log(N|\mathcal{Q}|)}{NB_1}}.$$

The proof is deferred to Appendix E, where we prove a stronger high-probability bound rather than a bound in expectation. From the theorem, we know that the cost of the finite-sample caching policy converges to the cost of the optimal policy at a rate of $1/\sqrt{N}$, which achieves the optimal dependence on $N$. The insights from the tabular case also indicate that the cost needs to be estimated in a conservative fashion when considered for the cache replacement algorithm.

---

[2]If we impose a uniform lower bound on the probability $P(q)$, then the pessimism can be replaced with the plug-in estimator. However, it is usually not the case in practice since $P(q)$ usually comes with a long tail.

### 3.3 Finite sample setting: Online learning

We summarize the caching algorithm pipeline in 1, which relies on the two estimation oracles, DenEstOracle and RegressionOracle, which estimate both the frequency and cost of models from data.

---

**Algorithm 1** Caching in Online Learning

---

1: Initialize the set of cache $\mathcal{L}_1 = \{\}$, past observations $\mathcal{H}_1 = \{\}$, $\hat{c}_{l,0}(q) = B_1, \forall q \in \mathcal{Q}$.
2: **For** iteration $t = 1, 2 \cdots, T$
3:     Receive query $q_t$.
4:     Update the density estimation $\hat{P}_t = \mathsf{DenEstOracle}(q_1, \cdots, q_t)$.
5:     **If** $q_t \in \mathcal{L}_t$:
6:         Output the cached result, set $\hat{c}_{l,t} = \hat{c}_{l,t-1}$, update the past observation $\mathcal{H}_t = \mathcal{H}_{t-1} \bigcup (q_t, \times)$, and continue.
7:     Use the large model to process the query, and observe a cost $c_t \sim \mathbb{P}_{C_l(q)}$.
8:     Update the past observation $\mathcal{H}_t = \mathcal{H}_{t-1} \bigcup (q_t, c_t)$.
9:     Update $\hat{c}_{l,t} = \mathsf{RegressionOracle}(\mathcal{H}_t)$.
10:    **If** $|\mathcal{L}_t| < L$:
11:       Let $\mathcal{L}_{t+1}$ be the union of $\mathcal{L}_t$ and $q_t$.
12:    **Else if** $\hat{P}_t(q_t) \cdot \hat{c}_{l,t}(q_t) > \min_{q \in \mathcal{L}_t} \hat{P}_t(q) \cdot \hat{c}_{l,t}(q)$ :
13:       Replace the minimizer element of $\hat{P}_t(q) \cdot \hat{c}_{l,t}(q)$ in the cache $\mathcal{L}_t$ with $q_t$ to get $\mathcal{L}_{t+1}$.

---

For theoretical analysis, we focus on the tabular case and define the oracles as follows:

$$\hat{P}_t(q) = \frac{\sum_{i=1}^t \mathbb{1}(q_i = q)}{t}, \tag{3}$$

$$\hat{c}_{l,t}(q) = \begin{cases} B_1, & \text{if } \sum_{i=1}^t \mathbb{1}(c_i \neq \times, q_i = q) = 0, \\ \max\left(B_1, \frac{\sum_{i=1}^t \mathbb{1}(c_i \neq \times, q_i = q)c_i}{\sum_{i=1}^t \mathbb{1}(c_i \neq \times, q_i = q)} - (B_2 - B_1)\sqrt{\frac{\log(6T|\mathcal{Q}|/\delta)}{2\sum_{i=1}^t \mathbb{1}(c_i \neq \times, q_i = q)}}\right), & \text{otherwise} \end{cases} \tag{4}$$

For the estimation of density, we use plug-in estimator since there is no imbalance in the sampling process. For the estimation of the cost, we subtract the confidence bound to include pessimism. We have the following regret guarantee.

**Theorem 2.** *When substituting the* DenEstOracle *and* RegressionOracle *with Equation (3) and (4) and set $\delta = 1/T$, we have for some universal constant $C$:*

$$\mathsf{Regret}_{\mathsf{cache}}(T) \leq \frac{CL(B_2 - B_1)B_2|\mathcal{Q}|L\log^2(T|\mathcal{Q}|)}{B_1} \cdot \sqrt{T}.$$

*On the other hand, for any caching policy $\{\mathcal{L}_t\}_{t=1}^T$, there exist some cases of $P(q), c_l^\star(q)$ such that for some universal constant $C'$,*

$$\mathsf{Regret}_{\mathsf{cache}}(T) \geq C'\sqrt{T}.$$

The proof is deferred to Appendix F. Different from the offline case, one interesting feature of the online case is the *partial observation phenomenon*: when the query hits the cache, it will not be processed by the model, and thus we cannot observe the sample from $C_l(q)$ in this round. This is different from the traditional bandit literature where the selected arm is always observed in each round. Thus the partial observation thus requires new upper and lower bound analysis.

## 4 Optimal Caching and Model multiplexing

### 4.1 Population setting

In the case when we have access to two models, we need to design a good caching and model multiplexing strategy jointly. We can compute the optimal caching and model multiplexing policy as

$\mathcal{L}^\star, \pi^\star = \arg\min_{\mathcal{L},\pi} \text{cost}(\mathcal{L}, \pi)$, which gives the following solution:

$$\pi^\star(q) = \mathbb{1}(c_s^\star(q) \leq c_l^\star(q)),$$

$$\mathcal{L}^\star = \arg\min_{\mathcal{L}:|\mathcal{L}|\leq L} \sum_{q\in\mathcal{Q}} P(q)\mathbb{1}(q \notin \mathcal{L})\min\left(c_s^\star(q), c_l^\star(q)\right).$$

Such optimal strategies are straightforward: $\pi^\star$ always assigns the query to the model with a smaller cost, and $\mathcal{L}^\star$ saves the $L$ queries with the largest $P(q) \cdot \min\left(c_s^\star(q), c_l^\star(q)\right)$.

For the model multiplexing algorithm, we consider two baselines: (a) one always uses large model $\pi_l(q) \equiv 0$; (b) one always uses the small model $\pi_s(q) \equiv 0$. This is related to the LLM cascade idea in the concurrent work of Chen et al. (2023b). We provide more discussion in Appendix A, and present comparisons between baselines and $\pi^\star$ in Appendix D.

## 4.2 Finite sample setting: Offline learning

We now consider the finite sample case. Let $\mathcal{D}_N = \{(q_1, c_{s,1}, c_{l,1}), \cdots, (q_N, c_{s,N}, c_{l,N})\}$, where $c_{s,n}$ is a sample from random variable $C_s(q_n)$, the observed cost for processing query $q_n$ with the small model in round $n$. And $c_{l,n}$ is a sample from random variable $C_l(q_n)$, the observed cost for processing query $q_n$ with the large model in round $n$. We consider estimating $P, c_s^\star, c_t^\star$ with some oracles $\hat{P} = \text{DenEstOracle}(q_1, \cdots, q_N)$, $\hat{c}_s(q), \hat{c}_t(q) = \text{RegressionOracle}(\mathcal{D}_N)$. We focus on the tabular case for theoretical analysis, where we set $\hat{P}$, $\hat{c}_s(q)$ and $\hat{c}_l(q)$ to be the plug-in estimator:

$$\hat{P}(q) = \frac{\sum_{i=1}^N \mathbb{1}(q_i = q)}{N}, \hat{c}_l(q) = \begin{cases} \frac{\sum_{i=1}^N \mathbb{1}(q_i=q)c_{l,i}}{\sum_{i=1}^N \mathbb{1}(q_i=q)}, & \text{if } \sum_{i=1}^N \mathbb{1}(q_i = q) > 0 \\ B_1, & \text{if } \sum_{i=1}^N \mathbb{1}(q_i = q) = 0, \end{cases}$$

$$\hat{c}_s(q) = \begin{cases} \frac{\sum_{i=1}^N \mathbb{1}(q_i=q)c_{s,i}}{\sum_{i=1}^N \mathbb{1}(q_i=q)}, & \text{if } \sum_{i=1}^N \mathbb{1}(q_i = q) > 0 \\ B_1, & \text{if } \sum_{i=1}^N \mathbb{1}(q_i = q) = 0. \end{cases}$$

Similar to the case of caching without model multiplexing, for a long-tailed distribution $P(q)$, the estimation of $c_s^\star(q), c_l^\star(q)$ can be bad for the queries that are visited less. To select the maximum $L$ elements from the plug-in estimator, we introduce pessimism to the estimate of $\hat{c}_l$ and $\hat{c}_s$. This leads to the following design of caching and model multiplexer $\hat{L}$ and $\hat{\pi}$:

$$\hat{\pi}(q) = \mathbb{1}(\hat{c}_s(q) \leq \hat{c}_l(q)),$$

$$\hat{\mathcal{L}} = \arg\min_{\mathcal{L}:|\mathcal{L}|\leq L} \sum_{q\in\mathcal{Q}} \mathbb{1}(q \notin \mathcal{L})\hat{P}(q) \max\left(B_1, \min(\hat{c}_s(q), \hat{c}_l(q)) - (B_2 - B_1)\sqrt{\frac{\log(8|\mathcal{Q}|/\delta)}{2\sum_{n=1}^N \mathbb{1}(q_n = q)}}\right).$$

We now show the cost for the caching and model multiplexer obtained from the empirical estimate is close to the optimal cost. The proof is deferred to Appendix G.

**Theorem 3.** *Assume that $N \geq \frac{8B_2|\mathcal{Q}|\log(4L/\delta)}{B_1}$ and take $\delta = 1/N$. We have*

$$\mathbb{E}[\text{cost}(\hat{\mathcal{L}}, \hat{\pi}) - \text{cost}(\mathcal{L}^\star, \pi^\star)] \leq CL(B_2 - B_1) \cdot \sqrt{\frac{B_2|\mathcal{Q}|\log(8|\mathcal{Q}|N)}{B_1 N}}.$$

## 4.3 Finite sample setting: Online learning

We turn to the online case. We first propose a meta-algorithm in Algorithm 2. We provide a theoretical analysis of the meta-algorithm for the tabular case, with $\text{DenEstOracle } \hat{P}_t(q) = \frac{\sum_{i=1}^t \mathbb{1}(q_i=q)}{t}$, and the RegressionOracle defined as follows:

$$\hat{c}_{l,t}(q) = \begin{cases} B_1, & \text{if } \sum_{i=1}^t \mathbb{1}(s_i = 0, q_i = q) = 0 \\ \max\left(B_1, \frac{\sum_{i=1}^t \mathbb{1}(s_i=0,q_i=q)c_{l,i}}{\sum_{i=1}^t \mathbb{1}(s_i=0,q_i=q)} - (B_2 - B_1)\sqrt{\frac{\log(8T|\mathcal{Q}|/\delta)}{2\sum_{i=1}^t \mathbb{1}(s_i=0,q_i=q)}}\right), & \text{otherwise,} \end{cases}$$

$$\hat{c}_{s,t}(q) = \begin{cases} B_1, & \text{if } \sum_{i=1}^t \mathbb{1}(s_i = 1, q_i = q) = 0, \\ \max\left(B_1, \frac{\sum_{i=1}^t \mathbb{1}(s_i=1,q_i=q)c_{s,i}}{\sum_{i=1}^t \mathbb{1}(s_i=1,q_i=q)} - (B_2 - B_1)\sqrt{\frac{\log(8T|\mathcal{Q}|/\delta)}{2\sum_{i=1}^t \mathbb{1}(s_i=1,q_i=q)}}\right), & \text{otherwise.} \end{cases}$$

We provide the following theorem on the regret of the overall algorithm.

**Algorithm 2** Joint Design of Caching and Model multiplexing

1: Initialize the set of cache $\mathcal{L}_1 = \{\}$, past observations $\mathcal{H}_1 = \{\}$, $\hat{c}_{l,0}(q) = B_1$, $\hat{c}_{s,0}(q) = B_1$, model multiplexing policy $\pi_0(q) = 1, \forall q \in \mathcal{Q}$.
2: **For** iteration $t = 1, 2 \cdots, T$
3:     Receive query $q_t$.
4:     Update the density estimation $\hat{P}_t = \mathsf{DenEstOracle}(q_1, \cdots, q_t)$.
5:     **If** $q_t \in \mathcal{L}_t$: output the cached result, set $\hat{c}_{s,t} = \hat{c}_{s,t-1}, \hat{c}_{l,t} = \hat{c}_{l,t-1}, \pi_t = \pi_{t-1}$, update the past observation $\mathcal{H}_t = \mathcal{H}_{t-1} \bigcup (q_t, \times, \times)$, and continue.
6:     Select the models according to $s_t = \pi_t(q_t)$.
7:     Update the past observation $\mathcal{H}_t = \mathcal{H}_{t-1} \bigcup (q_t, s_t, c_t)$.
8:     Update $\hat{c}_{l,t}, \hat{c}_{s,t} = \mathsf{RegressionOracle}(\mathcal{H}_t)$. Set $\pi_{t+1}(q) = \mathbb{1}(\hat{c}_{s,t}(q) < \hat{c}_{l,t}(q))$.
9:     **If** $|\mathcal{L}_t| < L$: let $\mathcal{L}_{t+1}$ be the union of $\mathcal{L}_t$ and $q_t$.
10:     **Else if** $\hat{P}_t(q_t) \cdot \min(\hat{c}_{s,t}(q_t), \hat{c}_{l,t}(q_t)) > \min_{q \in \mathcal{L}_t} \hat{P}_t(q) \cdot \min(\hat{c}_{s,t}(q), \hat{c}_{l,t}(q))$:
11:         replace the minimizer element in the cache $\mathcal{L}_t$ on the RHS with $q_t$ to get $\mathcal{L}_{t+1}$.

**Theorem 4.** *Substituting the oracles in Algorithm 2 with the oracles above and $\delta = 1/T$, we have*

$$\mathsf{Regret}_{\mathsf{sel}}(T) \leq \frac{CL(B_2 - B_1)B_2|\mathcal{Q}|L\log^2(T|\mathcal{Q}|)}{B_1} \cdot \sqrt{T}.$$

The proof is deferred to Appendix H. Compared with the lower bound in Theorem 2, we see that the dependency on $T$ is tight. The pessimism plays two different roles here: on the one hand, it encourages the exploration for model multiplexing to choose the ones with more uncertainty in the cost; on the other hand, it encourages the exploitation to be conservative about which query to save into the cache.

For the model multiplexer to work well, one needs to have a small yet accurate model multiplexer. In the case when the model multiplexer is not accurate, the small model always comes with a much smaller cost, and we are allowed to regenerate the responses and make corrections for the output, one may combine LEC with cascade (Chen et al., 2023b) to achieve better performance.

## 5 Experiments

We conduct both simulations and real-world experiments with our proposed methods. The code is available at `https://github.com/Ying1123/llm-caching-multiplexing`.

### 5.1 Simulations for algorithm analysis

We conduct synthetic online and offline experiments for joint optimization of caching and model switching. In Figure 2, we plot the cumulative cost and regret in online learning for LFU and LEC caching algorithms. For LFU, we consider model switchers which always select the small or large models as the baselines. We consider 20 distinct prompts and set the cache size to be 10. We set the frequency distribution as power distribution with $\alpha = 0.9$. The ground truth cost for each query processed by both models is set as a sample from $100X + 1$, where $X$ is a random variable generated from a Bernoulli distribution with the parameter $0.5$. We repeat the simulation 100 times and plot the mean and standard deviation in the figure. Our simulation suggests that LEC with model switcher greatly improves the two baselines by a factor of $50\times$ when the cost ratio is 100. We include additional results on the synthetic datasets for both online and offline settings with different $\alpha$ values, cost ratios, and switcher accuracy in Appendix I.1.

### 5.2 Experiments on real datasets

We evaluate our algorithms on two tasks: next-token prediction on the Lambada (Paperno et al., 2016) dataset and chat assistant on the OpenAssistant (Köpf et al., 2023) dataset.

For the next-token prediction task, we run the offline algorithm with two models: OPT-1.3B and OPT-13B (Zhang et al., 2022) and use FLOPs as the cost. The target performance metric is the number of correct tokens predicted, where we get the ground-truth token from the Lambada dataset.

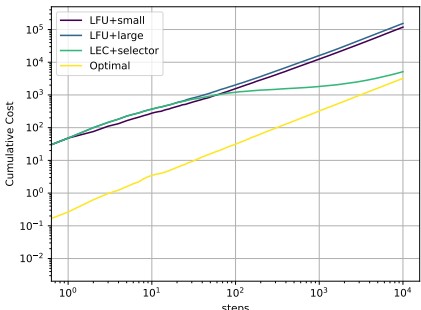 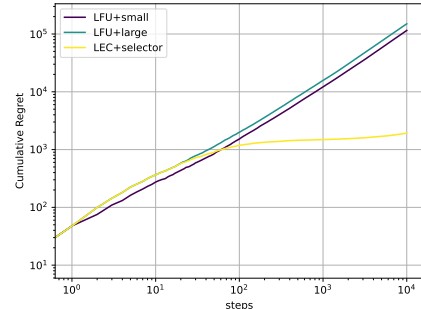

Figure 2: Comparisons between LFU with either small or large model switching and LEC with model switcher. Both the $x$-axis and $y$-axis are logarithmic scales. The shaded regime represents the standard deviation calculated from the repeated experiments.

For a given query, an algorithm can choose to run the small model or the large model. If the small model is chosen but its result is wrong, the large model must be run and it will incur an additional penalty. We fine-tune a BERT base model with $2000$ samples as the model switcher by predicting whether the small model can give the correct result and achieve 80.2% accuracy. We work with $100$ unseen distinct prompts in the offline setting with total queries $10000$ and cache size $40$. We compare our offline caching and switcher algorithms against LFU, large-model-only, and cascade (which always calls the small model first). As shown in Table 1, LEC is better than LFU in all cases. Combining LEC and switcher brings up to $4.3\times$ cost reduction compared to the baseline "LFU + Large." However, as the predictor accuracy is limited, the model switcher may not be as good as the cascade algorithm in some cases. We leave the training of a better switcher as future work.

On the chat assistant task, we run the online algorithm with two models: FastChat-T5-3B and Vicuna-13B (Chiang et al., 2023), and use the inference latency as the cost. The quality of response is evaluated by GPT4 evaluation (Liu et al., 2023). We say a response is satisfying if the score is larger than 6 out of 10, and unsatisfying otherwise. If the response from the small model is unsatisfying, we will call the large model again and incur an additional cost in latency. The ratio between the average latency of the large model and the small model is 1.85. We work with $100$ distinct prompts in the online setting with total queries $10000$ and cache size $40$. After a sufficient number of online learning steps, the switcher learns the accurate costs of two models on this finite prompts set, so "LEC + switcher" outperforms other algorithms in all cases on Table 2 with up to $1.8\times$ latency reduction compared to "LFU + large" baseline.

| $\alpha$ | selector accuracy | LFU+ large | LFU+ cascade | LFU+ selector | LEC+ large | LEC+ cascade | LEC+ selector |
|---|---|---|---|---|---|---|---|
| 0.2 | 80% | 3.49 | 3.81 | 2.60 | 3.44 | **1.50** | 2.00 |
| 0.8 | 80% | 10.81 | 11.80 | 8.06 | 10.36 | **4.11** | 4.76 |
| 0.2 | 100% | 3.49 | 3.81 | 1.91 | 3.44 | 1.50 | **0.99** |
| 0.8 | 100% | 10.81 | 11.80 | 5.90 | 10.36 | 4.11 | **2.50** |

Table 1: Evaluation of offline algorithms on the Lambada dataset with OPT-1.3B and OPT-13B, $100$ distinct prompts, total query size $10000$ and cache size $40$. $\alpha$ is the parameter of the power distribution of the prompts. The table lists cumulative costs ($10^3$) for different algorithms.

We provide more experiments in Appendix I, where we evaluate both FLOPs and latency for both offline and online setting on both synthetic and real dataset, with varying cache size, query size and distinct prompts.

| $\alpha$ | LFU+ large | LFU+ cascade | LFU+ selector | LEC+ large | LEC+ cascade | LEC+ selector |
|---|---|---|---|---|---|---|
| 0.2 | 9.31 | 13.88 | 7.24 | 8.74 | 8.82 | **5.93** |
| 0.5 | 20.04 | 29.88 | 15.11 | 18.68 | 16.90 | **11.87** |
| 0.8 | 28.24 | 42.12 | 21.14 | 26.07 | 20.31 | **15.49** |

Table 2: Evaluation of online algorithms on the OpenAssistant dataset with FastChat-T5-3B and Vicuna-13B, 100 distinct prompts, total query size 10000 and cache size 40. $\alpha$ is the parameter of the power distribution of the prompts. The table lists cumulative costs ($10^3$) for different algorithms.

## 6  Conclusions

We have studied the joint optimization of caching and model multiplexing and proposed an optimal algorithm for the tabular case. There are a variety of further work that can be pursued in this vein, including:

- Designing the optimal caching and model multiplexing algorithm when there is a query queue, such that the query arrives at a random interval rather than a fixed interval. A more complicated serving pattern also needs to take batching strategies into consideration.
- Understanding the scaling law of the predictors. We hope to use a small yet accurate model for prediction to reduce overhead introduced by the predictor. It is important to understand the trade-off between prediction accuracy, model size, and training data size.
- Designing optimal caching algorithm when the responses generated in each round have diverse qualities.

## Acknowledgements

Banghua and Jiantao are partially supported by NSF IIS-1901252, CIF-1909499 and CIF-2211209. Michael I. Jordan is partially supported by NSF IIS-1901252. Ying Sheng and Clark Barrett are supported in part by NSF-2110397 and the Stanford Center for Automated Reasoning.

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

# Appendix

## A   Discussions on the Choice of Output, Model and Cost

The proposed framework is flexible in the choice of outputs, models and costs. Below we discuss several possible choices and combinations of output, models and costs that are most practically relevant.

**Per-token Output and Per-sentence Output.**   We have two design choices of the desired output in each round, namely per-token output and per-sentence output.

For per-token output, we aim at generating one token at each round as a response of the queries. In this case, we only cache the next token for a given query and estimate the cost for generating next token. We also have the flexibility of choosing different models to generate each token in each round.

For per-sentence output, we aim at generating a complete response at each round. In this case, we cache the whole responses for a given query, and estimate the cost for generating the whole responses. This may introduce more variance in the cost due to the variation and randomness in the length of the generated responses.

**Choices of Costs**   The cost can be chosen as FLOPS, latency of the model, the price for API calls, user satisfaction of the results, or a combination of all the four factors.

**Model multiplexing**   A common choice of model ensembles is a pair of small and large models. The cost for small model $C_s(q)$ can be written as $C_s(q) = C_{s,0}(q) + Y(q)C_{s,1}(q)$. Here $Y(q)$ is a binary random variable, indicating whether the small model outputs satisfying results ($Y(q) = 0$) or not ($Y(q) = 1$). In the case when the small model outputs a satisfying response, the incurred cost is $C_{s,0}(q)$. In the case when the small model outputs a bad response, the incurred cost is $C_{s,0}(q) + C_{s,1}(q)$. We discuss two possible choices of $Y(q)$, $C_{s,0}(q)$ and $C_{s,1}(q)$ based on two different evaluation pipeline as below.

- **One-time evaluation pipeline.**   For the one-time evaluation pipeline, we can only call one of the models once and the generated content cannot be changed. In this case, $C_{s,0}(q)$ can be set as the cost for running the small model to generate responses, $Y(q)$ is set to be 1 if the user is not satisfied with the response, and $C_{s,1}(q)$ is the incurred cost for unsatisfactory of the user. One can similarly set the same cost for the large model.

- **Correction-based evaluation pipeline**.   For correction-based evaluation, we may re-generate the content with a different model if it is unsatisfying, and get an extra cost for fixing the content. Such evaluation can be easily combined with LLM Cascade (Frantar et al., 2023) or the idea from Big Little Transformer Decoder (Kim et al., 2023) and Spectulative sampling (Chen et al., 2023a). For example, after running the small model, we run the large model once to infer all the log probabilities of the small model output in parallel, and reject its output if the log probabilities are low. If the small model output is rejected, we will set $Y(q) = 1$ and run large model to re-generate the responses. In this case, $C_{s,0}(q)$ is the cost of running the small model for generating responses, and running the large model once for checking the probability. And $C_{s,1}(q)$ is the cost of running the large model to generate the response.

We also remark here that in the special case when the cost for the small model is much smaller than that of the large model under the correction-based evaluation pipeline, the cascade selector which always runs the small model first may give better performance than the model multiplexer if the accuracy of the model multiplexer is low, since running small model does not introduce too much cost compared to running large model. In this situation, the cascade selector can also be combined with LEC caching to further improve the performance.

On the other hand, we may also choose among models with similar size but different expertise, including coding, summarization and chat etc. In this case, we also expect to see different qualities and cost of responses for specific queries.

## B Generalization to Variable Size Cache

For the variable-size caching problem, assume that the cache size of $q$ is a deterministic scalar, denoted as $S(q)$. In the population case we design the cache as follows:

$$\mathcal{L}^\star = \underset{\mathcal{L}:\sum_{q\in\mathcal{L}} S(q)\leq L}{\arg\min} \sum_{q\in\mathcal{Q}} P(q)\mathbb{1}(q\notin\mathcal{L})\min\left(c_s^\star(q), c_l^\star(q)\right).$$

In the case when all $S, P, c_s^\star, c_l^\star$ are known, one may solve the above constrained optimization problem for the optimal caching. When $S(q) \ll L$, a good cache replacement algorithm is GDSF itself, which replaces the query with the smallest expected cost per-size $P(q)\min\left(c_s^\star(q), c_l^\star(q)\right)/S(q)$ rather than expected cost per-query.

A more practical setting is the case when the cache size for each query $S(q)$ is a random variable. Due to the randomness in the generation procedure, we expect to see responses of different lengths even when we use the same model to process the same query. In each round, we will have a generated response with size $s(q)$ that is sampled from the random variable $S(q_t)$. We conjecture that the optimal cache replacement algorithm is to replace the query with the smallest expected cost per-size $P(q)\min\left(c_s^\star(q), c_l^\star(q)\right)/s(q)$ as well, where $s(q)$ is the size of the cached queries and responses.

## C Generalization to Multiplexing of Multiple Models

The proposed algorithm can be generalized to model multiplexing with multiple models. Assume that we have $K$ models, and each model has a random cost function $C_k(q)$ with expectation $c_k^\star(q)$. In this case, the optimal population algorithm is

$$\pi^\star(q) = \underset{k\in[K]}{\arg\min}\, c_k^\star(q),$$

$$\mathcal{L}^\star = \underset{\mathcal{L}:|\mathcal{L}|\leq L}{\arg\min} \sum_{q\in\mathcal{Q}} P(q)\mathbb{1}(q\notin\mathcal{L})\min_{k\in[K]} c_k^\star(q).$$

And the finite sample algorithm is natural to follow. In practice, one may train a neural network with $K$ dimensional output to predict the cost for each of the models.

## D Differences Between the Optimal Policy and the Baseline

Consider the population setting in Section 3, where we optimize caching without model multiplexing. We show via a simple example below that without considering the cost for individual query, LFU can be highly sub-optimal compared to the optimal caching strategy in the population. The ratio

**Proposition 1.** *For any fixed cost function $c_l^\star$, one can design some distribution of queries $P$ such that for any $\epsilon > 0$,*

$$\frac{\mathsf{cost}(\mathcal{L}_{\mathsf{LFU}})}{\mathsf{cost}(\mathcal{L}_{\mathsf{LEC}})} \geq \frac{\max_{q\in\mathcal{Q}} c_l^\star(q)}{\min_{q\in\mathcal{Q}} c_l^\star(q)} - \epsilon.$$

The construction can be seen from a two-query example. Let $c_l^\star(q_1) = c_1, c_l^\star(q_2) = c_2$ with $c_1 < c_2$. Let $P(q_1) =, P(q_2) =$ This shows that when the individual cost varies drastically for different queries, the total expected cost for LFU can be highly sub-optimal compared with the cost-aware caching strategy.

To compare the performance of the model multiplexing in Section 4, we take the cache size $L = 0$. We have the following proposition for the performance improvement of the model multiplexer.

**Proposition 2.** *Let $L = 0$. The difference in cost between the baseline and the model multiplexer can be written as*

$$\mathsf{cost}(\mathcal{L}^\star, \pi_s) - \mathsf{cost}(\mathcal{L}^\star, \pi^\star) = \sum_{q\in\mathcal{Q}} P(q)\max(0, c_s^\star(q) - c_l^\star(q)),$$

$$\mathsf{cost}(\mathcal{L}^\star, \pi_l) - \mathsf{cost}(\mathcal{L}^\star, \pi^\star) = \sum_{q\in\mathcal{Q}} P(q)\max(0, c_l^\star(q) - c_s^\star(q)).$$

The proof is a direct result of plugging in the cost definition. We see that the gap between $\pi_s$ and the optimal model multiplexer becomes larger when a large fraction of the queries have smaller cost when processed by the large models, and vice versa.

## E   Proof of Theorem 1

*Proof.* We first prove the following lemma on the lower bound of $P(q)$ for any $q \in \mathcal{L}^\star$.

**Lemma 1.** *For any $q \in \mathcal{L}^\star$, we have $P(q) \geq B_1/(B_2|\mathcal{Q}|)$.*

*Proof.* From the fact that $\sum_{q \in \mathcal{Q}} P(q) = 1$ and for any $q \in \mathcal{L}^\star$ and any $q' \notin \mathcal{L}^\star$, $P(q) \geq P(q')c_l^\star(q')/c_l^\star(q) \geq P(q')B_1/B_2$, we know that for any $q \in \mathcal{L}^\star$, $P(q) \geq B_1/(B_2|\mathcal{Q}|)$.  □

We define the following three events:

$$E_1 = \left\{ \forall q \in \mathcal{Q}, |\hat{P}(q) - P(q)| \leq \sqrt{\frac{2\log(6/\delta)}{N}} \right\},$$

$$E_2 = \left\{ \forall q \in \mathcal{Q}, |\hat{c}_l(q) - c_l^\star(q)| \leq (B_2 - B_1)\sqrt{\frac{\log(6|\mathcal{Q}|/\delta)}{2\sum_{n=1}^N \mathbb{1}(q_n = q)}} \right\},$$

$$E_3 = \left\{ \forall q \in \mathcal{L}^\star, \sum_{n=1}^N \mathbb{1}(q_n = q) \geq \frac{B_1 N}{2B_2|\mathcal{Q}|} \right\}.$$

We know that the first two events hold simutaneously with probability at least $1 - 2\delta/3$ from Lemma 3. For the third event, from the Chernoff bound, we know that for any $q \in \mathcal{Q}$, we have

$$\mathbb{P}\left( \sum_{n=1}^N \mathbb{1}(q_n = q) \geq NP(q)/2 \right) \geq 1 - \exp(-NP(q)/8).$$

From Lemma 1 we know that for any $q \in \mathcal{L}^\star$, $P(q) \geq B_1/(B_2|\mathcal{Q}|)$. Thus the above inequality further implies

$$\mathbb{P}\left( \sum_{n=1}^N \mathbb{1}(q_n = q) \geq \frac{B_1 N}{2B_2|\mathcal{Q}|} \right) \geq 1 - \exp\left( -\frac{B_1 N}{8B_2|\mathcal{Q}|} \right) \geq 1 - \frac{\delta}{3L}.$$

The last inequality is due to our assumption that $N \geq \frac{8B_2|\mathcal{Q}|\log(3L/\delta)}{B_1}$.

We condition on the three events from now on. The last two events imply that for any $q \in \mathcal{L}^\star$,

$$|\hat{c}_l(q) - c_l^\star(q)| \leq (B_2 - B_1)\sqrt{\frac{B_2|\mathcal{Q}|\log(6|\mathcal{Q}|/\delta)}{NB_1}}.$$

We have

$$\mathsf{cost}(\hat{\mathcal{L}}) - \mathsf{cost}(\mathcal{L}^\star) = \sum_{q \in \mathcal{Q}} P(q)\left( \mathbb{1}(q \notin \hat{\mathcal{L}}) - \mathbb{1}(q \notin \mathcal{L}^\star) \right) c_l^\star(q)$$

$$= \sum_{q \in \mathcal{Q}} P(q)\left( \mathbb{1}(q \in \mathcal{L}^\star) - \mathbb{1}(q \in \hat{\mathcal{L}}) \right) c_l^\star(q).$$

Let $\hat{c}_{l,pes}(q) = \hat{c}_l(q) - (B_2 - B_1)\sqrt{\frac{\log(6|\mathcal{Q}|/\delta)}{2\sum_{n=1}^N \mathbb{1}(q_n = q)}}$. Note that for any $q \in \mathcal{L}^\star$, we know that

$$\hat{P}(q)\hat{c}_{l,pes}(q) \geq \max\left( P(q) - \sqrt{\frac{2\log(6/\delta)}{N}}, 0 \right)\left( c_l^\star(q) - 2(B_2 - B_1)\sqrt{\frac{B_2|\mathcal{Q}|\log(6|\mathcal{Q}|/\delta)}{NB_1}} \right)$$

$$\geq P(q)c_l^\star(q) - C(B_2 - B_1) \cdot \sqrt{\frac{B_2|\mathcal{Q}|\log(6|\mathcal{Q}|/\delta)}{NB_1}}.$$

And similarly, for any $q \notin \mathcal{L}^\star$, we know that

$$\hat{P}(q)\hat{c}_{l,pes}(q) \leq \left( P(q) + \sqrt{\frac{2\log(6/\delta)}{N}} \right) c_l^\star(q) \leq P(q)c_l^\star(q) + B_2\sqrt{\frac{2\log(6/\delta)}{N}}.$$

Now consider any $q \in \hat{\mathcal{L}}$ but $q \notin \mathcal{L}^\star$, and any other $q' \in \mathcal{L}^\star$ but $q' \notin \hat{\mathcal{L}}$. We have

$$P(q')c_l^\star(q') - P(q)c_l^\star(q)$$

$$\leq \hat{P}(q')\hat{c}_{l,pes}(q') - \hat{P}(q)\hat{c}_{l,pes}(q) + C(B_2 - B_1) \cdot \sqrt{\frac{B_2|\mathcal{Q}|\log(6|\mathcal{Q}|/\delta)}{NB_1}}$$

$$\leq C(B_2 - B_1) \cdot \sqrt{\frac{B_2|\mathcal{Q}|\log(6|\mathcal{Q}|/\delta)}{NB_1}}.$$

Overall, we know that conditioned on $E_1 \cap E_2 \cap E_3$, we have

$$\mathsf{cost}(\hat{\mathcal{L}}) - \mathsf{cost}(\mathcal{L}^\star) \leq C(B_2 - B_1)L \cdot \sqrt{\frac{B_2|\mathcal{Q}|\log(6|\mathcal{Q}|/\delta)}{NB_1}}.$$

And this implies that

$$\mathbb{E}[\mathsf{cost}(\hat{\mathcal{L}}) - \mathsf{cost}(\mathcal{L}^\star)] \leq C(B_2 - B_1)L \cdot \sqrt{\frac{B_2|\mathcal{Q}|\log(6|\mathcal{Q}|/\delta)}{NB_1}} + \delta B_2.$$

Taking $\delta = 1/N$ finishes the proof. $\qquad\square$

## F  Proof of Theorem 2

*Proof.* **Upper Bound.** We start with the upper bound by the following lemma.

**Lemma 2.** *In each round $t \in [T]$, we always have*

$$\mathcal{L}_{t+1} \in \arg\min_{\mathcal{L}} \sum_{q \in \mathcal{Q}} \hat{P}_t(q)\mathbb{1}(q \notin \mathcal{L})\hat{c}_{l,t}(q).$$

*Proof.* We prove this lemma by induction. First, consider the case when $|\mathcal{L}_{t+1}| < L$. In this scenario, we always put the query into the cache. And $\mathcal{L}_{t+1}$ contains all queries with non-zero $\hat{P}_t$. Thus such $\mathcal{L}_{t+1}$ is always one of the minimizers.

Now consider the case when $|\mathcal{L}_{t+1}| = L$. Assume that the conclusion holds for time step $t$. Now consider the case of $t + 2$. When the new query is in the cache $q_{t+1} \in \mathcal{L}_{t+1}$, the cache will remain unchanged $\mathcal{L}_{t+2} = \mathcal{L}_{t+1}$. In this case, the estimated probability for $q_{t+1}$ is increased, while the others are decreased, and $\hat{c}_{l,t+1}$ is not changed for any query. Thus $\mathcal{L}_{t+2}$ is still the minimizer. When the new query does not hit the cache, the estimated probability times costs for all other queries except for $q_{t+1}$ are decreased proportionally since $\hat{P}_{t+1}$ is decreased proportionally while $\hat{c}_{l,t+1}$ is not changed for all other queries. Thus the only potential change in the relative order of costs is that of $q_{t+1}$. Since we can add $q_{t+1}$ at the end of query, we know that after this round $\mathcal{L}_{t+2}$ is still the minimizer. $\qquad\square$

Let $g_k(q)$ be the length of the interval between the $k$-th and $k+1$-th arrival of query $q$ in the sequence of received queries (we set $g_k(q) = 0$ if $k$ exceeds the total number of times $q$ is queried.). Define the following three events:

$$E_{1,t} = \left\{ \forall q \in \mathcal{Q}, |\hat{P}_{t-1}(q) - P(q)| \leq \min\left( 1, \sqrt{\frac{2\log(6T/\delta)}{t-1}} \right) \right\},$$

$$E_{2,t} = \left\{ \forall q \in \mathcal{Q}, \hat{c}_{l,t-1}(q) - c_l^\star(q) \in \left[ -2(B_2 - B_1)\min\left( 1, \sqrt{\frac{\log(6T|\mathcal{Q}|/\delta)}{2\sum_{i=1}^{t-1}\mathbb{1}(c_i \neq \times, q_i = q)}} \right), 0 \right] \right\},$$

$$E_3 = \left\{ \forall q \in \mathcal{L}^\star, k \leq T, g_k(q) \leq \frac{B_2|\mathcal{Q}|\log(3TL/\delta)}{B_1} \right\}.$$

We prove that the three events hold simultaneously with probability at least $1 - \delta$:

**Lemma 3.** *We have*

$$\mathbb{P}\left(\left(\bigcap_{t=T^{2/3}}^{T} E_{1,t} \cap E_{2,t}\right) \cap E_3\right) \geq 1 - \delta.$$

*Proof.* From the Dvoretzky-Kiefer-Wolfowitz inequality, we have

$$\mathbb{P}(\max_{q\in\mathcal{Q}} |\hat{P}_t(q) - P(q)| > \epsilon) \leq 2\exp(-\epsilon^2 t/2).$$

By taking $\epsilon = \sqrt{\frac{2\log(6T/\delta)}{t}}$, we see that $\max_{q\in\mathcal{Q}} |\hat{P}_t(q) - P(q)| \leq \epsilon$ holds with probability at least $1 - \delta/(3T)$ for any fixed $t \in [T]$. Now by taking a union bound over all $t \in [T]$, we know that $\bigcap_{t=1}^{T} E_{1,t}$ holds with probability at least $1 - \delta/3$.

For the second event, from Hoeffding's inequality, we have for any $q \in \mathcal{Q}$,

$$\mathbb{P}\left(|\hat{c}_{l,t}(q) - c_l^\star(q)| \leq (B_2 - B_1)\min\left(1, \sqrt{\frac{\log(6T|\mathcal{Q}|/\delta)}{2\sum_{s=1}^{t-1} \mathbb{1}(c_s \neq \times, q_s = q)}}\right)\right) \geq 1 - \frac{\delta}{3T|\mathcal{Q}|}.$$

Now taking union bound over $t \in [T]$ and $q \in \mathcal{Q}$ gives that $\bigcap_{t=1}^{T} E_{2,t}$ holds with probability at least $1 - \delta/3$.

For the third event, we know that the interval $g_k(q)$ satisfies a geometric distribution with success probability $P(q)$. For any $q \in \mathcal{L}^\star$, we have

$$\mathbb{P}\left(g_k(q) \geq s\right) \leq (1 - P(q))^s \leq \left(1 - \frac{B_1}{|\mathcal{Q}|B_2}\right)^s.$$

By taking $s = \frac{B_2|\mathcal{Q}|\log(TL/\delta)}{B_1}$, we know that

$$\mathbb{P}\left(g_k(q) \geq \frac{B_2|\mathcal{Q}|\log(3TL/\delta)}{B_1}\right) \leq \left(1 - \frac{B_1}{|\mathcal{Q}|B_2}\right)^s \leq \frac{\delta}{3TL}.$$

By taking union bounds over all $q \in \mathcal{L}^\star$ and $k$ we get the result. $\qquad\square$

Let $E^t = \bigcap_{s=1}^{t} E_{1,s} \cap E_{2,s}$. We can write the regret as follows.

$$\text{Regret}(T) \leq \sum_{t=1}^{T} \mathbb{E}[\text{cost}(q_t, \mathcal{L}_t) - \text{cost}(q_t, \mathcal{L}^\star)\mathbb{1}(E^t)] + \mathbb{E}[\text{cost}(q_t, \mathcal{L}_t) - \text{cost}(q_t, \mathcal{L}^\star)\mathbb{1}(\bar{E}^t)]$$

$$\leq \sum_{t=1}^{T} \mathbb{E}[\text{cost}(q_t, \mathcal{L}_t) - \text{cost}(q_t, \mathcal{L}^\star)\mathbb{1}(E^t)] + C\delta T B_2$$

$$= C\delta T B_2 + \sum_{t=1}^{T} \mathbb{E}[\text{cost}(q_t, \mathcal{L}_t) - \text{cost}(q_t, \mathcal{L}^\star)\mathbb{1}(E^t)].$$

Note that the sampling distribution of $q_t$ is independent of $E^t$. Thus we can write the expectation as

$$\sum_{t=1}^{T} \mathbb{E}[\text{cost}(q_t, \mathcal{L}_t) - \text{cost}(q_t, \mathcal{L}^\star)\mathbb{1}(E^t)] \leq \sum_{t=1}^{T}\sum_{q\in\mathcal{Q}} \mathbb{E}[P(q)\left(\mathbb{1}(q \notin \mathcal{L}_t) - \mathbb{1}(q \notin \mathcal{L}^\star)\right)c_l^\star(q) \mid E^t]$$

Let $T_t(q) = \sum_{i=1}^{t-1} \mathbb{1}(q_i \notin \mathcal{L}_i, q_i = q)$. Note that the event $c_i = \times$ is equivalent to that $q_i \in \mathcal{L}_i$. Now at each round $t$, conditioned on event $E^t$, we know that for any $q \in \mathcal{L}^\star$,

$$\hat{P}_{t-1}(q)\hat{c}_{l,t-1}(q) \geq \max\left(P(q) - \min\left(1, \sqrt{\frac{2\log(6T/\delta)}{t-1}}\right), 0\right)\left(c_l^\star(q) - 2(B_2 - B_1)\cdot\min\left(1, \sqrt{\frac{\log(6T|\mathcal{Q}|/\delta)}{T_t(q)}}\right)\right)$$

$$\geq P(q)c_l^\star(q) - C(B_2 - B_1)\cdot\min\left(1, \sqrt{\frac{\log(6T|\mathcal{Q}|/\delta)}{T_t(q)}}\right).$$

And similarly, for any $q \notin \mathcal{L}^\star$, we know that

$$\hat{P}_t(q)\hat{c}_{l,t-1}(q) \leq \left(P(q) + \min\left(1, \sqrt{\frac{2\log(8T/\delta)}{t-1}}\right)\right)c_l^\star(q) \leq P(q)c_l^\star(q) + B_2\min\left(1, \sqrt{\frac{2\log(8T/\delta)}{t-1}}\right).$$

Now consider any $q \in \mathcal{L}_t$ but $q \notin \mathcal{L}^\star$, and any other $q' \in \mathcal{L}^\star$ but $q' \notin \mathcal{L}_t$. We have

$$P(q')c_l^\star(q') - P(q)c_l^\star(q)$$
$$\leq \hat{P}(q')\hat{c}_{l,t-1}(q') - \hat{P}(q)\hat{c}_{l,t-1}(q) + C(B_2 - B_1) \cdot \min\left(1, \sqrt{\frac{\log(6T|\mathcal{Q}|/\delta)}{T_t(q)}}\right) + B_2\min\left(1, \sqrt{\frac{2\log(6T/\delta)}{t-1}}\right)$$
$$\leq C(B_2 - B_1) \cdot \min\left(1, \sqrt{\frac{\log(6T|\mathcal{Q}|/\delta)}{T_t(q)}}\right) + B_2\min\left(1, \sqrt{\frac{2\log(8T/\delta)}{t-1}}\right).$$

Thus we know that

$$\sum_{t=1}^{T} \mathbb{E}[\mathsf{cost}(q_t, \mathcal{L}_t) - \mathsf{cost}(q_t, \mathcal{L}^\star)\mathbb{1}(E^t)]$$
$$\leq C\sum_{t=1}^{T}\mathbb{E}\left[\sum_{q\in\mathcal{L}^\star}\mathbb{1}(q \notin \mathcal{L}_t)(B_2 - B_1) \cdot \min\left(1, \sqrt{\frac{\log(6T|\mathcal{Q}|/\delta)}{T_t(q)}}\right) + B_2\min\left(1, \sqrt{\frac{2\log(6T/\delta)}{t-1}}\right) \mid E^t\right].$$

Thus we have

$$\sum_{t=1}^{T} \mathbb{E}[\mathsf{cost}(q_t, \mathcal{L}_t) - \mathsf{cost}(q_t, \mathcal{L}^\star)\mathbb{1}(E^t)]$$
$$\leq B_2T\delta + C\sum_{t=1}^{T}\mathbb{E}\left[\sum_{q\in\mathcal{L}^\star}\mathbb{1}(q \notin \mathcal{L}_t)(B_2 - B_1) \cdot \min\left(1, \sqrt{\frac{\log(6T|\mathcal{Q}|/\delta)}{T_t(q)}}\right)\right.$$
$$\left. + B_2\min\left(1, \sqrt{\frac{2\log(6T/\delta)}{t-1}}\right) \mid E^t \cap E_3\right]$$
$$\leq C \cdot \left(B_2T\delta + LB_2\sqrt{2T\log(6T/\delta)}\right.$$
$$\left. + (B_2 - B_1)\log(6T|\mathcal{Q}|/\delta) \cdot \sum_{q\in\mathcal{L}^\star}\sum_{t=1}^{T}\mathbb{E}\left[\mathbb{1}(q \notin \mathcal{L}_t) \cdot \min\left(1, \sqrt{\frac{1}{T_t(q)}}\right) \mid E^t \cap E_3\right]\right).$$

Now for each $q \in \mathcal{L}^\star$, we look at the term $\sum_{t=1}^{T}\mathbb{E}\left[\mathbb{1}(q \notin \mathcal{L}_t) \cdot \min\left(1, \sqrt{\frac{1}{T_t(q)}}\right) \mid E^t \cap E_3\right]$. We prove the following lemma:

**Lemma 4.** *We have*

$$\sum_{t=1}^{T}\mathbb{E}\left[\mathbb{1}(q \notin \mathcal{L}_t) \cdot \min\left(1, \sqrt{\frac{1}{T_t(q)}}\right) \mid E^t \cap E_3\right] \leq \frac{CB_2|\mathcal{Q}|\log(3TL/\delta)\sqrt{T}}{B_1} + T\delta.$$

*Proof.* Let $t_k(q) = \sum_{l=1}^{k-1} g_l(q)$ be the step that the $k$-th query of $q$ arrives, with $t_0(q) = 0$. And let $E = (\bigcap_{t=1}^{T} E_t) \cap E_3$. The summation can be written as

$$\sum_{t=1}^{T} \mathbb{E}\left[ \mathbb{1}(q \notin \mathcal{L}_t) \cdot \min\left(1, \sqrt{\frac{1}{T_t(q)}}\right) \mid E^t \cap E_3 \right]$$

$$\leq \sum_{t=1}^{T} \mathbb{E}\left[ \mathbb{1}(q \notin \mathcal{L}_t) \cdot \min\left(1, \sqrt{\frac{1}{T_t(q)}}\right) \mid E \right] + T\delta$$

$$= \sum_{k=0}^{T} \mathbb{E}\left[ \sum_{t=t_k(q)+1}^{t_{k+1}(q)} \mathbb{1}(q \notin \mathcal{L}_t) \cdot \min\left(1, \sqrt{\frac{1}{T_t(q)}}\right) \mid E \right] + T\delta$$

$$\leq \sum_{k=0}^{T} \mathbb{E}\left[ \sum_{t=t_k(q)+1}^{t_{k+1}(q)} \mathbb{1}(q \notin \mathcal{L}_{t_{k+1}(q)}) \cdot \min\left(1, \sqrt{\frac{1}{T_{t_k(q)+1}(q)}}\right) \mid E \right] + T\delta.$$

The last inequality is due to (a) $T_t(q)$ does not change if at round $t$ the query is not $q$; (b) if $q \in \mathcal{L}_{t_{k+1}(q)}$, we will have $q \in \mathcal{L}_t$ for any $t \in [t_k(q)+1, t_{k+1}(q)]$ since $q$ never arrives in the middle and must remain in the cache set until $t_{k+1}(q)$. Now from event $E_3$, we know that

$$\sum_{k=0}^{T} \mathbb{E}\left[ \sum_{t=t_k(q)+1}^{t_{k+1}(q)} \mathbb{1}(q \notin \mathcal{L}_{t_{k+1}(q)}) \cdot \min\left(1, \sqrt{\frac{1}{T_{t_k(q)+1}(q)}}\right) \mid E \right]$$

$$\leq \sum_{k=0}^{T} \mathbb{E}\left[ \mathbb{1}(q \notin \mathcal{L}_{t_{k+1}(q)}) \cdot g_k(q) \cdot \min\left(1, \sqrt{\frac{1}{T_{t_k(q)+1}(q)}}\right) \mid E \right]$$

$$\leq \frac{B_2 |\mathcal{Q}| \log(3TL/\delta)}{B_1} \cdot \sum_{k=0}^{T} \mathbb{E}\left[ \mathbb{1}(q \notin \mathcal{L}_{t_{k+1}(q)}) \cdot \min\left(1, \sqrt{\frac{1}{T_{t_k(q)+1}(q)}}\right) \mid E \right]$$

We know that $T_{t_{k+1}(q)+1}(q) = T_{t_{k+1}(q)}(q) + 1 = T_{t_k(q)+1}(q) + 1$ if $q \notin \mathcal{L}_{t_{k+1}(q)}$ since the query $q$ missing the cache will be sent to the model. Thus overall, we know that we have either $T_{t_{k+1}(q)+1}(q) = T_{t_k(q)+1}(q) + 1$, or $\mathbb{1}(q \notin \mathcal{L}_{t_{k+1}(q)}) \cdot \sqrt{\frac{1}{T_{t_k(q)+1}(q)}} = 0$ and $T_{t_{k+1}(q)+1}(q) = T_{t_k(q)+1}(q)$. Thus overall, we have

$$\sum_{k=0}^{T} \mathbb{E}\left[ \mathbb{1}(q \notin \mathcal{L}_{t_{k+1}(q)}) \cdot \min\left(1, \sqrt{\frac{1}{T_{t_k(q)+1}(q)}}\right) \mid E \right] \leq \sum_{k=1}^{T} \frac{1}{\sqrt{k}} \leq C\sqrt{T}.$$

$\square$

By taking $\delta = 1/T$, we know the final regret can be bounded by

$$\mathsf{Regret}(T) \leq \frac{CL(B_2 - B_1)B_2|\mathcal{Q}|L\log^2(T|\mathcal{Q}|)}{B_1} \cdot \sqrt{T}.$$

**Lower bound.** Now we turn to the lower bound. We apply Le Cam's two point lemma for the regret. Consider any family of algorithm $\{\mathcal{L}_t\}_{t=1}^{T}$, where $\mathcal{L}_t$ can be dependent on observations prior to time step $t$. We aim to design two instances with the same $P(q)$ and different random variable $C_l(q)$ such that for any algorithm, the incurred cost for one of the instance is at least $\Omega(\sqrt{T})$. Consider the case when we only have two candidate queries $\mathcal{Q} = \{q_1, q_2\}$. Set $P(q_1) = P(q_2) = 1/2$ for both instances and the cache size $L = 1$. For instance one, we let $C_l^{(1)}(q_1) \sim \mathsf{Bern}(1/2)$, $C_l^{(1)}(q_2) \sim \mathsf{Bern}(1/2 + \Delta)$. For instance two, we let $C_l^{(2)}(q_1) \sim \mathsf{Bern}(1/2)$, $C_l^{(2)}(q_2) \sim \mathsf{Bern}(1/2 - \Delta)$. We have

$$\inf_{\{\mathcal{L}_t\}_{t=1}^{T}} \sup_{P, C_l} \mathsf{Regret}(T) \geq \inf_{\{\mathcal{L}_t\}_{t=1}^{T}} \sup_{C_l \in \{C_l^{(1)}, C_l^{(2)}\}} \mathsf{Regret}(T)$$

$$= \inf_{\{\mathcal{L}_t\}_{t=1}^{T}} \sup_{C_l \in \{C_l^{(1)}, C_l^{(2)}\}} \sum_{t=1}^{T} \mathbb{E}\left[ \frac{1}{2} \sum_{i=1}^{2} \mathbb{1}(q_i \notin \mathcal{L}_t) c_l^{\star}(q_i) - \frac{1}{2} \sum_{i=1}^{2} \mathbb{1}(q_i \notin \mathcal{L}^{\star}) c_l^{\star}(q_i) \right].$$

Let $\mathsf{Regret}^{(1)}(T)$ be the total regret when $C_l = C_l^{(1)}$, and $\mathsf{Regret}^{(2)}(T)$ be the total regret when $C_l = C_l^{(2)}$. Then we can verify that for any sequence of $\mathcal{L}_t$,

$$\mathsf{Regret}^{(1)}(T) + \mathsf{Regret}^{(2)}(T) \geq \frac{\Delta T}{2}.$$

Thus from Le Cam's Lemma, we have

$$\begin{aligned}
\inf_{\{\mathcal{L}_t\}_{t=1}^T} \sup_{P, C_l} \mathsf{Regret}(T) &\geq \frac{\Delta T}{4} \cdot (1 - \mathsf{TV}(\mathbb{P}_{c_l^{(1)}}, \mathbb{P}_{c_l^{(2)}})) \\
&\geq \frac{\Delta T}{8} \cdot \exp(-D_{\mathsf{KL}}(\mathbb{P}_{c_l^{(1)}}, \mathbb{P}_{c_l^{(2)}})) \\
&\geq \frac{\Delta T}{8} \cdot \exp(-2\Delta^2 \mathbb{E}_1[T_2]).
\end{aligned}$$

Here $\mathbb{E}_1[T_2]$ is the expected times of observing the cost of $q_2$ under instance one. Taking $\Delta = T^{-1/2}$ and minimizing the above equation with $\mathbb{E}_1[T_2]$ gives the desired bound. $\qquad\square$

## G  Proof of Theorem 3

*Proof.* We define the following four events:

$$E_1 = \left\{ \forall q \in \mathcal{Q}, |\hat{P}(q) - P(q)| \leq \sqrt{\frac{2\log(8/\delta)}{N}} \right\},$$

$$E_2 = \left\{ \forall q \in \mathcal{Q}, |\hat{c}_l(q) - c_l^\star(q)| \leq (B_2 - B_1)\sqrt{\frac{\log(8|\mathcal{Q}|/\delta)}{2\sum_{n=1}^N \mathbb{1}(q_n = q)}} \right\},$$

$$E_3 = \left\{ \forall q \in \mathcal{Q}, |\hat{c}_s(q) - c_s^\star(q)| \leq (B_2 - B_1)\sqrt{\frac{\log(8|\mathcal{Q}|/\delta)}{2\sum_{n=1}^N \mathbb{1}(q_n = q)}} \right\},$$

$$E_4 = \left\{ \forall q \in \mathcal{L}^\star, \sum_{n=1}^N \mathbb{1}(q_n = q) \geq N \cdot P(q)/2 \right\}.$$

We know that the above events hold simultaneously with probability at least $1 - \delta$ from Lemma 3. We condition on the four events from now on. We first decompose the cost difference as

$$\mathsf{cost}(\hat{\mathcal{L}}, \hat{\pi}) - \mathsf{cost}(\mathcal{L}^\star, \pi^\star) = \mathsf{cost}(\hat{\mathcal{L}}, \hat{\pi}) - \mathsf{cost}(\hat{\mathcal{L}}, \pi^\star) + \mathsf{cost}(\hat{\mathcal{L}}, \pi^\star) - \mathsf{cost}(\mathcal{L}^\star, \pi^\star).$$

The first difference can be further written as

$$\begin{aligned}
\mathsf{cost}(\hat{\mathcal{L}}, \hat{\pi}) - \mathsf{cost}(\hat{\mathcal{L}}, \pi^\star) &= \sum_{q \in \mathcal{Q}} P(q)\mathbb{1}(q \notin \hat{\mathcal{L}})(c_s^\star(q)\hat{\pi}(q) + c_l^\star(q)(1 - \hat{\pi}(q)) - c_s^\star(q)\pi^\star(q) - c_l^\star(q)(1 - \pi^\star(q))) \\
&= \sum_{q \in \mathcal{Q}} P(q)\mathbb{1}(q \notin \hat{\mathcal{L}})(c_s^\star(q)\hat{\pi}(q) + c_l^\star(q)(1 - \hat{\pi}(q)) - \min(c_s^\star(q), c_l^\star(q))) \\
&\leq \sum_{q \in \mathcal{Q}} P(q)(c_s^\star(q)\hat{\pi}(q) + c_l^\star(q)(1 - \hat{\pi}(q)) - \min(c_s^\star(q), c_l^\star(q))) \\
&= \sum_{q \in \mathcal{Q}} P(q) \left( c_s^\star(q)\mathbb{1}(\hat{c}_s(q) \leq \hat{c}_l(q)) + c_l^\star(q)\mathbb{1}(\hat{c}_s(q) > \hat{c}_l(q)) - \min(c_s^\star(q), c_l^\star(q)) \right).
\end{aligned}$$

Note that if $\hat{c}_s(q) - \hat{c}_l(q)$ has the same sign as $c_s^\star(q) - c_l^\star(q)$, the difference $c_s^\star(q)\mathbb{1}(\hat{c}_s(q) \leq \hat{c}_l(q)) + c_l^\star(q)\mathbb{1}(\hat{c}_s(q) > \hat{c}_l(q)) - \min(c_s^\star(q), c_l^\star(q))$ becomes 0. Otherwise, if $c_s^\star(q) - c_l^\star(q) > 0$, we know that

$$c_s^\star(q) - c_l^\star(q) \leq \hat{c}_s(q) - \hat{c}_l(q) + |\hat{c}_s(q) - c_s^\star(q)| + |\hat{c}_l(q) - c_l^\star(q)| \leq |\hat{c}_s(q) - c_s^\star(q)| + |\hat{c}_l(q) - c_l^\star(q)|.$$

And similarly if $c_s^\star(q) - c_l^\star(q) \leq 0$, we know that $c_l^\star(q) - c_s^\star(q) \leq |\hat{c}_s(q) - c_s^\star(q)| + |\hat{c}_l(q) - c_l^\star(q)|$. Overall, we have

$$\mathbb{E}[\mathsf{cost}(\hat{\mathcal{L}}, \hat{\pi}) - \mathsf{cost}(\hat{\mathcal{L}}, \pi^\star)] \leq \mathbb{E}\left[\sum_{q \in \mathcal{Q}} P(q)|\hat{c}_s(q) - c_s^\star(q)| + |\hat{c}_l(q) - c_l^\star(q)|\right]$$

$$\overset{(i)}{\leq} \mathbb{E}\left[\sqrt{\sum_{q \in \mathcal{Q}} P(q)(\hat{c}_s(q) - c_s^\star(q))^2} + \sqrt{\sum_{q \in \mathcal{Q}} P(q)(\hat{c}_l(q) - c_l^\star(q))^2}\right]$$

$$\overset{(ii)}{\leq} \sqrt{\mathbb{E}\left[\sum_{q \in \mathcal{Q}} P(q)(\hat{c}_s(q) - c_s^\star(q))^2\right]} + \sqrt{\mathbb{E}\left[\sum_{q \in \mathcal{Q}} P(q)(\hat{c}_l(q) - c_l^\star(q))^2\right]}$$

$$\overset{(iii)}{\leq} C(B_2 - B_1)\sqrt{\frac{|\mathcal{Q}|\log(N)}{N}}.$$

Here $(i)$ is due to Cauchy-Schwarz, and (ii) is from Jensen's inequality, and (iii) is the standard rate of the least squared estimator (Rigollet and Hütter, 2015).

For the second difference, we have

$$\mathsf{cost}(\hat{\mathcal{L}}, \pi^\star) - \mathsf{cost}(\mathcal{L}^\star, \pi^\star) = \sum_{q \in \mathcal{Q}} P(q)\left(\mathbb{1}(q \notin \hat{\mathcal{L}}) - \mathbb{1}(q \notin \mathcal{L}^\star)\right)\min(c_s^\star(q), c_l^\star(q))$$

$$= \sum_{q \in \mathcal{Q}} P(q)\left(\mathbb{1}(q \in \mathcal{L}^\star) - \mathbb{1}(q \in \hat{\mathcal{L}})\right)\min(c_s^\star(q), c_l^\star(q)).$$

Note that for any $q \in \mathcal{L}^\star$, we know that

$$\hat{P}(q)\left(\min(\hat{c}_s(q), \hat{c}_l(q)) - (B_2 - B_1)\sqrt{\frac{\log(8|\mathcal{Q}|/\delta)}{2\sum_{n=1}^{N}\mathbb{1}(q_n = q)}}\right)$$

$$\geq \max\left(P(q) - \sqrt{\frac{2\log(8/\delta)}{N}}, 0\right) \cdot \left(\min(c_s^\star(q), c_l^\star(q)) - 2(B_2 - B_1)\sqrt{\frac{\log(8|\mathcal{Q}|/\delta)}{2\sum_{n=1}^{N}\mathbb{1}(q_n = q)}}\right)$$

$$\geq P(q)\min(c_s^\star(q), c_l^\star(q)) - C(B_2 - B_1) \cdot \sqrt{\frac{\log(8|\mathcal{Q}|/\delta)}{2\sum_{n=1}^{N}\mathbb{1}(q_n = q)}}$$

$$\geq P(q)\min(c_s^\star(q), c_l^\star(q)) - C(B_2 - B_1) \cdot \sqrt{\frac{B_2|\mathcal{Q}|\log(8|\mathcal{Q}|/\delta)}{B_1 N}}.$$

The last inequality uses event $E_3$ and Lemma 1. And similarly, for any $q \notin \mathcal{L}^\star$, we know that

$$\hat{P}(q)\left(\min(\hat{c}_s(q), \hat{c}_l(q)) - (B_2 - B_1)\sqrt{\frac{\log(8|\mathcal{Q}|/\delta)}{2\sum_{n=1}^{N}\mathbb{1}(q_n = q)}}\right) \leq \left(P(q) + \sqrt{\frac{2\log(8/\delta)}{N}}\right)\min(c_s^\star(q), c_l^\star(q))$$

$$\leq P(q)\min(c_s^\star(q), c_l^\star(q)) + B_2\sqrt{\frac{2\log(8/\delta)}{N}}.$$

Now consider any $q \in \mathcal{L}_t$ but $q \notin \mathcal{L}^\star$, and any other $q' \in \mathcal{L}^\star$ but $q' \notin \mathcal{L}_t$. We have

$$P(q')\min(c_s^\star(q'), c_l^\star(q')) - P(q)\min(c_s^\star(q), c_l^\star(q))$$

$$\leq \hat{P}(q')\left(\min(\hat{c}_s(q'), \hat{c}_l(q')) - (B_2 - B_1)\sqrt{\frac{\log(8|\mathcal{Q}|/\delta)}{2\sum_{n=1}^{N}\mathbb{1}(q_n = q')}}\right)$$

$$- \hat{P}(q)\left(\min(\hat{c}_s(q), \hat{c}_l(q)) - (B_2 - B_1)\sqrt{\frac{\log(8|\mathcal{Q}|/\delta)}{2\sum_{n=1}^{N}\mathbb{1}(q_n = q)}}\right) + C(B_2 - B_1) \cdot \sqrt{\frac{B_2|\mathcal{Q}|\log(8|\mathcal{Q}|/\delta)}{B_1 N}}$$

$$\leq C(B_2 - B_1) \cdot \sqrt{\frac{B_2|\mathcal{Q}|\log(8|\mathcal{Q}|/\delta)}{B_1 N}}.$$

Here the last inequality uses the fact that $q$ is inside $\mathcal{L}_t$ and thus the difference between the first two terms are upper bounded by 0. Finally, we know that conditioned on $E_1 \cap E_2 \cap E_3 \cap E_4$, we have

$$\text{cost}(\hat{\mathcal{L}}, \pi^\star) - \text{cost}(\mathcal{L}^\star, \pi^\star) \leq CL(B_2 - B_1) \cdot \sqrt{\frac{B_2|\mathcal{Q}|\log(8|\mathcal{Q}|/\delta)}{B_1 N}}.$$

Overall, we know that

$$\mathbb{E}[\text{cost}(\hat{\mathcal{L}}, \hat{\pi}) - \text{cost}(\hat{\mathcal{L}}, \pi^\star)] \leq B_2\delta + CL(B_2 - B_1) \cdot \sqrt{\frac{B_2|\mathcal{Q}|\log(8|\mathcal{Q}|/\delta)}{B_1 N}}.$$

Taking $\delta = 1/N$ finishes the proof. $\qquad\square$

## H   Proof of Theorem 4

*Proof.* Let $g_k(q)$ be the length of the interval between the $k$-th and $(k+1)$-th arrival of query $q$ in the sequence of received queries (we set $g_k(q) = 0$ if $k$ exceeds the total number of times $q$ is queried.). Define the following four events:

$$E_{1,t} = \left\{ \forall q \in \mathcal{Q}, |\hat{P}_{t-1}(q) - P(q)| \leq \min\left(1, \sqrt{\frac{2\log(8T/\delta)}{t-1}}\right) \right\},$$

$$E_{2,t} = \left\{ \forall q \in \mathcal{Q}, \hat{c}_{l,t-1}(q) \in \left[c_l^\star(q) - 2(B_2 - B_1)\min\left(1, \sqrt{\frac{\log(8T|\mathcal{Q}|/\delta)}{2\sum_{i=1}^{t-1}\mathbb{1}(s_i = 0, q_i = q)}}\right), c_l^\star(q)\right] \right\},$$

$$E_{3,t} = \left\{ \forall q \in \mathcal{Q}, \hat{c}_{s,t-1}(q) \in \left[c_s^\star(q) - 2(B_2 - B_1)\min\left(1, \sqrt{\frac{\log(8T|\mathcal{Q}|/\delta)}{2\sum_{i=1}^{t-1}\mathbb{1}(s_i = 1, q_i = q)}}\right), c_s^\star(q)\right] \right\},$$

$$E_4 = \left\{ \forall q \in \mathcal{L}^\star, k \leq T, g_k(q) \leq \frac{B_2|\mathcal{Q}|\log(4TL/\delta)}{B_1} \right\}.$$

From the same analysis as Lemma 3, we know that the four events $(\bigcap_{s=1}^{t} E_{1,s} \cap E_{2,s} \cap E_{3,s}) \cap E_4$ hold simultaneously with probability at least $1 - \delta$.

Let $E^t = \bigcap_{s=1}^{t} E_{1,s} \cap E_{2,s} \cap E_{3,s}$. The regret can be decomposed as follows.

$\text{Regret}(T)$

$$= \sum_{t=1}^{T} \mathbb{E}[\text{cost}(q_t, \mathcal{L}_t, \pi_t) - \text{cost}(q_t, \mathcal{L}^\star, \pi^\star)]$$

$$\leq \sum_{t=1}^{T} \mathbb{E}[\text{cost}(q_t, \mathcal{L}_t, \pi_t) - \text{cost}(q_t, \mathcal{L}^\star, \pi^\star)\mathbb{1}(E^t)] + \mathbb{E}[\text{cost}(q_t, \mathcal{L}_t, \pi_t) - \text{cost}(q_t, \mathcal{L}^\star, \pi^\star)\mathbb{1}(\bar{E}^t)]$$

$$\leq \sum_{t=1}^{T} \mathbb{E}[\text{cost}(q_t, \mathcal{L}_t, \pi_t) - \text{cost}(q_t, \mathcal{L}^\star, \pi^\star)\mathbb{1}(E^t)] + \delta T B_2$$

$$= \sum_{t=1}^{T} \mathbb{E}[(\text{cost}(q_t, \mathcal{L}_t, \pi_t) - \text{cost}(q_t, \mathcal{L}_t, \pi^\star) + \text{cost}(q_t, \mathcal{L}_t, \pi^\star) - \text{cost}(q_t, \mathcal{L}^\star, \pi^\star))\mathbb{1}(E^t)] + \delta T B_2.$$

The first difference can be further written as

$$\mathbb{E}[\text{cost}(q_t, \mathcal{L}_t, \pi_t) - \text{cost}(q_t, \mathcal{L}_t, \pi^\star) \mid E^t]$$
$$= \mathbb{E}[\mathbb{1}(q_t \notin \mathcal{L}_t)(c_s^\star(q_t)\pi_t(q_t) + c_l^\star(q_t)(1 - \pi_t(q_t)) - c_s^\star(q_t)\pi^\star(q_t) - c_l^\star(q_t)(1 - \pi^\star(q_t))) \mid E^t]$$
$$= \mathbb{E}[\mathbb{1}(q_t \notin \mathcal{L}_t)(c_s^\star(q_t)\pi_t(q_t) + c_l^\star(q_t)(1 - \pi_t(q_t)) - \min(c_s^\star(q_t), c_l^\star(q_t))) \mid E^t]$$
$$\leq \mathbb{E}[c_s^\star(q_t)\pi_t(q_t) + c_l^\star(q_t)(1 - \pi_t(q_t)) - \min(c_s^\star(q_t), c_l^\star(q_t)) \mid E^t]$$
$$= \mathbb{E}[c_s^\star(q_t)\mathbb{1}(\hat{c}_{s,t}(q_t) \leq \hat{c}_{l,t}(q_t)) + c_l^\star(q_t)\mathbb{1}(\hat{c}_{s,t}(q_t) > \hat{c}_{l,t}(q_t)) - \min(c_s^\star(q_t), c_l^\star(q_t)) \mid E^t].$$

Note that if $\hat{c}_{s,t}(q_t) - \hat{c}_{l,t}(q_t)$ has the same sign as $c_s^\star(q_t) - c_l^\star(q_t)$, the difference $c_s^\star(q_t)\mathbb{1}(\hat{c}_{s,t}(q_t) \leq \hat{c}_{l,t}(q_t)) + c_l^\star(q_t)\mathbb{1}(\hat{c}_{s,t}(q_t) > \hat{c}_{l,t}(q_t)) - \min(c_s^\star(q_t), c_l^\star(q_t))$ becomes 0. Otherwise, if $c_s^\star(q_t) - c_l^\star(q_t) > 0$ and $\hat{c}_{s,t}(q_t) - \hat{c}_{l,t}(q_t) \leq 0$, we know that $s_t = 1$ and

$$c_s^\star(q) - c_l^\star(q) \leq \hat{c}_{s,t}(q) - \hat{c}_{l,t}(q) + 2(B_2 - B_1)\min\left(1, \sqrt{\frac{\log(8T|\mathcal{Q}|/\delta)}{2\sum_{i=1}^{t-1}\mathbb{1}(s_i = 1, q_i = q)}}\right)$$

$$\leq 2(B_2 - B_1)\min\left(1, \sqrt{\frac{\log(8T|\mathcal{Q}|/\delta)}{2\sum_{i=1}^{t-1}\mathbb{1}(s_i = 1, q_i = q)}}\right).$$

And similarly if $c_s^\star(q) - c_l^\star(q) \leq 0$ and $\hat{c}_{s,t}(q_t) - \hat{c}_{l,t}(q_t) > 0$, we know that $s_t = 0$ and $c_l^\star(q) - c_s^\star(q) \leq 2(B_2 - B_1)\min\left(1, \sqrt{\frac{\log(8T|\mathcal{Q}|/\delta)}{2\sum_{i=1}^{t-1}\mathbb{1}(s_i=0,q_i=q)}}\right)$. Overall, we have

$$\sum_{t=1}^{T}\mathbb{E}[(\text{cost}(q_t, \mathcal{L}_t, \pi_t) - \text{cost}(q_t, \mathcal{L}_t, \pi^\star)]$$

$$\leq 2(B_2 - B_1)\sum_{t=1}^{T}\sum_{q \in \mathcal{Q}}\mathbb{E}\Bigg[\mathbb{1}(s_t = 1, q_t = q)\min\left(1, \sqrt{\frac{\log(8T|\mathcal{Q}|/\delta)}{2\sum_{i=1}^{t-1}\mathbb{1}(s_i = 1, q_i = q)}}\right)$$

$$+ \mathbb{1}(s_t = 0, q_t = q)\min\left(1, \sqrt{\frac{\log(8T|\mathcal{Q}|/\delta)}{2\sum_{i=1}^{t-1}\mathbb{1}(s_i = 0, q_i = q)}}\right)\Bigg]$$

$$\leq 2(B_2 - B_1)\sqrt{|\mathcal{Q}|T\log(8|\mathcal{Q}|T/\delta)}.$$

Here the last inequality uses the fact that for each $q \in \mathcal{Q}$, the summation over time step is upper bounded by $2\sum_{i=1}^{T(q)}\sqrt{\log(8T|\mathcal{Q}|/\delta)/2i} \leq 2\sqrt{\log(8T|\mathcal{Q}|/\delta)T(q)}$, where $T(q)$ is the number of steps of receiving query $q$ in total $T$ steps. Optimizing over $T(q)$ gives the final bound.

For the second difference, we have

$$\mathbb{E}[\text{cost}(\mathcal{L}_t, \pi^\star) - \text{cost}(\mathcal{L}^\star, \pi^\star) \mid E^t] = \mathbb{E}\left[\sum_{q \in \mathcal{Q}}P(q)\left(\mathbb{1}(q \notin \mathcal{L}_t) - \mathbb{1}(q \notin \mathcal{L}^\star)\right)\min(c_s^\star(q), c_l^\star(q)) \mid E^t\right]$$

$$= \mathbb{E}\left[\sum_{q \in \mathcal{Q}}P(q)\left(\mathbb{1}(q \in \mathcal{L}^\star) - \mathbb{1}(q \in \mathcal{L}_t)\right)\min(c_s^\star(q), c_l^\star(q)) \mid E^t\right].$$

Note that for any $q \in \mathcal{L}^\star$, we know that

$$\hat{P}_t(q)\min\left(\hat{c}_{s,t}(q), \hat{c}_{l,t}(q)\right)$$

$$\geq \max\left(P(q) - \sqrt{\frac{2\log(8/\delta)}{t}}, 0\right) \cdot \mathbb{1}(\pi_t(q) = 1)\left(c_s^\star(q) - 2(B_2 - B_1)\min\left(1, \sqrt{\frac{\log(8|\mathcal{Q}|/\delta)}{2\sum_{i=1}^{t-1}\mathbb{1}(s_i = 1, q_i = q)}}\right)\right)$$

$$+ \mathbb{1}(\pi_t(q) = 0)\left(c_l^\star(q) - 2(B_2 - B_1)\min\left(1, \sqrt{\frac{\log(8|\mathcal{Q}|/\delta)}{2\sum_{i=1}^{t-1}\mathbb{1}(s_i = 0, q_i = q)}}\right)\right)$$

$$\geq P(q)\min\left(c_s^\star(q), c_l^\star(q)\right) - (B_2 - B_1)\sqrt{\frac{2\log(8/\delta)}{t}} + P(q) \cdot \Bigg(\mathbb{1}(\pi_t(q) = 1)$$

$$\cdot \left(c_s^\star(q) - 2(B_2 - B_1)\min\left(1, \sqrt{\frac{\log(8|\mathcal{Q}|/\delta)}{2\sum_{i=1}^{t-1}\mathbb{1}(s_i = 1, q_i = q)}}\right)\right)$$

$$+ \mathbb{1}(\pi_t(q) = 0)\left(c_l^\star(q) - 2(B_2 - B_1)\min\left(1, \sqrt{\frac{\log(8|\mathcal{Q}|/\delta)}{2\sum_{i=1}^{t-1}\mathbb{1}(s_i = 0, q_i = q)}}\right)\right) - \min(c_s^\star(q), c_l^\star(q))\Bigg)$$

$$\geq P(q)\min\left(c_s^\star(q), c_l^\star(q)\right) - C(B_2 - B_1) \cdot \min\left(1, \sqrt{\frac{\log(8|\mathcal{Q}|/\delta)}{2\sum_{i=1}^{t-1}\mathbb{1}(s_i = \pi_t(q), q_i = q)}}\right).$$

Below we justify the last inequality. First, note that $\pi_t(q) = 1$ is equivalent to that $\hat{c}_{s,t}(q) \leq \hat{c}_{l,t}(q)$. Thus if $\hat{c}_{s,t}(q_t) - \hat{c}_{l,t}(q_t)$ has the same sign as $c_s^\star(q_t) - c_l^\star(q_t)$, the above inequality holds. Now consider the case when $\hat{c}_{s,t}(q_t) - \hat{c}_{l,t}(q_t)$ has a different sign as $c_s^\star(q_t) - c_l^\star(q_t)$. Assume that $\hat{c}_{s,t}(q_t) > \hat{c}_{l,t}(q_t)$ and $c_s^\star(q_t) < c_l^\star(q_t)$. We know that $\pi_t(q) = 0$, and

$$c_l^\star(q) - 2(B_2 - B_1) \min\left(1, \sqrt{\frac{\log(8|\mathcal{Q}|/\delta)}{2\sum_{i=1}^{t-1} \mathbb{1}(s_i = 0, q_i = q)}}\right) - c_s^\star(q)$$

$$> -2(B_2 - B_1) \min\left(1, \sqrt{\frac{\log(8|\mathcal{Q}|/\delta)}{2\sum_{i=1}^{t-1} \mathbb{1}(s_i = 0, q_i = q)}}\right).$$

Similarly we can prove that for the reversed case. Now for any $q \notin \mathcal{L}^\star$, we know that

$$\hat{P}_t(q) \min\left(\hat{c}_{s,t}(q), \hat{c}_{l,t}(q)\right) \leq \left(P(q) + \sqrt{\frac{2\log(8/\delta)}{t}}\right) \min\left(c_s^\star(q), c_l^\star(q)\right)$$

$$\leq P(q) \min\left(c_s^\star(q), c_l^\star(q)\right) + B_2\sqrt{\frac{2\log(8/\delta)}{t}}.$$

Now consider any $q \in \mathcal{L}_t$ but $q \notin \mathcal{L}^\star$, and any other $q' \in \mathcal{L}^\star$ but $q' \notin \mathcal{L}_t$. We have

$$P(q') \min\left(c_s^\star(q'), c_l^\star(q')\right) - P(q) \min\left(c_s^\star(q), c_l^\star(q)\right)$$

$$\leq \hat{P}_t(q') \min\left(\hat{c}_{s,t}(q'), \hat{c}_{l,t}(q')\right) - \hat{P}_t(q) \min\left(\hat{c}_{s,t}(q), \hat{c}_{l,t}(q)\right)$$

$$+ C(B_2 - B_1) \cdot \min\left(1, \sqrt{\frac{\log(8|\mathcal{Q}|/\delta)}{2\sum_{i=1}^{t-1} \mathbb{1}(s_i = \pi_t(q), q_i = q)}}\right) + B_2\sqrt{\frac{2\log(8/\delta)}{t}}$$

$$\leq C(B_2 - B_1) \cdot \min\left(1, \sqrt{\frac{\log(8|\mathcal{Q}|/\delta)}{2\sum_{i=1}^{t-1} \mathbb{1}(s_i = \pi_t(q), q_i = q)}}\right) + B_2\sqrt{\frac{2\log(8/\delta)}{t}}.$$

Here the last inequality uses the fact that $q$ is inside $\mathcal{L}_t$. Thus we have

$$\sum_{t=1}^{T} \mathbb{E}[\text{cost}(\mathcal{L}_t, \pi^\star) - \text{cost}(\mathcal{L}^\star, \pi^\star)\mathbb{1}(E^t)]$$

$$\leq \sum_{t=1}^{T} \mathbb{E}[\text{cost}(\mathcal{L}_t, \pi^\star) - \text{cost}(\mathcal{L}^\star, \pi^\star)\mathbb{1}(E^t \cap E_4)] + B_2 T\delta$$

$$\leq B_2 T\delta + C\sum_{t=1}^{T} \mathbb{E}\left[\sum_{q \in \mathcal{L}^\star} \mathbb{1}(q \notin \mathcal{L}_t)(B_2 - B_1) \cdot \min\left(1, \sqrt{\frac{\log(8|\mathcal{Q}|/\delta)}{2\sum_{i=1}^{t-1} \mathbb{1}(s_i = \pi_t(q), q_i = q)}}\right)\right.$$

$$\left. + B_2\sqrt{\frac{2\log(8/\delta)}{t}} \mid E^t \cap E_4\right]$$

$$\leq C \cdot \left(B_2 T\delta + LB_2\sqrt{2T\log(8/\delta)} + (B_2 - B_1)\log(8T|\mathcal{Q}|/\delta)\right.$$

$$\cdot \sum_{q \in \mathcal{L}^\star} \sum_{t=1}^{T} \mathbb{E}\left[\mathbb{1}(q \notin \mathcal{L}_t) \cdot \min\left(1, \sqrt{\frac{1}{\sum_{i=1}^{t-1} \mathbb{1}(s_i = \pi_t(q), q_i = q)}}\right) \mid E^t \cap E_4\right]\right)$$

$$= C \cdot \left(B_2 T\delta + LB_2\sqrt{2T\log(8/\delta)} + (B_2 - B_1)\log(8T|\mathcal{Q}|/\delta)\right.$$

$$\left. \cdot \sum_{q \in \mathcal{L}^\star} \sum_{\pi \in \{1,2\}} \sum_{t=1}^{T} \mathbb{E}\left[\mathbb{1}(q \notin \mathcal{L}_t, \pi_t(q) = \pi) \cdot \min\left(1, \sqrt{\frac{1}{\sum_{i=1}^{t-1} \mathbb{1}(s_i = \pi, q_i = q)}}\right) \mid E^t \cap E_4\right]\right).$$

Let $T_t(q, \pi) = \sum_{i=1}^{t-1} \mathbb{1}(s_i = \pi, q_i = q)$. Now for each $q \in \mathcal{L}^\star$ and $\pi \in \{0, 1\}$, we look at the term $\sum_{t=1}^{T} \mathbb{E}\left[\mathbb{1}(q \notin \mathcal{L}_t, \pi_t = \pi) \cdot \min\left(1, \sqrt{\frac{1}{T_t(q,\pi)}}\right) \mid E^t \cap E_4\right]$. We prove the following lemma:

**Lemma 5.** *We have*

$$\sum_{t=1}^{T} \mathbb{E}\left[\mathbb{1}(q \notin \mathcal{L}_t, \pi_t(q) = \pi) \cdot \min\left(1, \sqrt{\frac{1}{T_t(q,\pi)}}\right) \mid E^t \cap E_4\right] \leq \frac{CB_2|\mathcal{Q}|\log(3TL/\delta)\sqrt{T}}{B_1} + T\delta.$$

*Proof.* Let $t_k(q) = \sum_{l=1}^{k-1} g_l(q)$ be the step that the $k$-th query of $q$ arrives. And let $E = (\bigcap_{t=1}^{T} E_t) \cap E_4$. The summation can be written as

$$\sum_{t=1}^{T} \mathbb{E}\left[\mathbb{1}(q \notin \mathcal{L}_t, \pi_t(q) = \pi) \cdot \min\left(1, \sqrt{\frac{1}{T_t(q,\pi)}}\right) \mid E^t \cap E_3\right]$$

$$\leq \sum_{t=1}^{T} \mathbb{E}\left[\mathbb{1}(q \notin \mathcal{L}_t, \pi_t(q) = \pi) \cdot \min\left(1, \sqrt{\frac{1}{T_t(q,\pi)}}\right) \mid E\right] + T\delta$$

$$= \sum_{k=0}^{T} \mathbb{E}\left[\sum_{t=t_k(q)+1}^{t_{k+1}(q)} \mathbb{1}(q \notin \mathcal{L}_t, \pi_t(q) = \pi) \cdot \min\left(1, \sqrt{\frac{1}{T_t(q,\pi)}}\right) \mid E\right] + T\delta$$

$$\leq \sum_{k=0}^{T} \mathbb{E}\left[\sum_{t=t_k(q)+1}^{t_{k+1}(q)} \mathbb{1}(q \notin \mathcal{L}_{t_{k+1}(q)}, \pi_{t_{k+1}}(q) = \pi) \cdot \min\left(1, \sqrt{\frac{1}{T_{t_k(q)+1}(q,\pi)}}\right) \mid E\right] + T\delta.$$

The last inequality is due to (a). $T_t(q,\pi)$ does not change if at round $t$ the query is not $q$; (b). if $q \in \mathcal{L}_{t_{k+1}(q)}$, we will have $q \in \mathcal{L}_t$ for any $t \in [t_k(q)+1, t_{k+1}(q)]$ since $q$ never arrives in the middle and must remain in the cache set until $t_{k+1}(q)$; (c) For $t \in [t_k(q)+1, t_{k+1}(q)]$, $\pi_t$ does not change since both the frequency and cost estimator does not change for $q$. From the definition of $t_k(q)$, we know that $T_{t_k(q)+1}(q,\pi) \geq 1$ and thus we can drop the the minimum in the above equation. Now from event $E_4$, we know that

$$\sum_{k=0}^{T} \mathbb{E}\left[\sum_{t=t_k(q)+1}^{t_{k+1}(q)} \mathbb{1}(q \notin \mathcal{L}_{t_{k+1}(q)}, \pi_{t_{k+1}}(q) = \pi) \cdot \min\left(1, \sqrt{\frac{1}{T_{t_k(q)+1}(q,\pi)}}\right) \mid E\right]$$

$$\leq \sum_{k=0}^{T} \mathbb{E}\left[g_k(q) \cdot \mathbb{1}(q \notin \mathcal{L}_{t_{k+1}(q)}, \pi_{t_{k+1}}(q) = \pi) \cdot \min\left(1, \sqrt{\frac{1}{T_{t_k(q)+1}(q,\pi)}}\right) \mid E\right]$$

$$\leq \frac{B_2|\mathcal{Q}|\log(4TL/\delta)}{B_1} \cdot \sum_{k=0}^{T} \mathbb{E}\left[\mathbb{1}(q \notin \mathcal{L}_{t_{k+1}(q)}, \pi_{t_{k+1}}(q) = \pi) \cdot \min\left(1, \sqrt{\frac{1}{T_{t_k(q)+1}(q,\pi)}}\right) \mid E\right]$$

We know that $T_{t_{k+1}(q)+1}(q,\pi) = T_{t_{k+1}(q)}(q,\pi) + 1 = T_{t_k(q)+1}(q,\pi) + 1$ if $q \notin \mathcal{L}_{t_{k+1}(q)}$ and $\pi_{t_{k+1}}(q) = \pi$ since the query $q$ missing the cache will be sent to one of the models, and only the one selected will observe the cost. Thus overall, we know that we have either $T_{t_{k+1}(q)+1}(q,\pi) = T_{t_k(q)+1}(q,\pi) + 1$, or $\mathbb{1}(q \notin \mathcal{L}_{t_{k+1}(q)}, \pi_{t_{k+1}}(q) = \pi) \cdot \min\left(1, \sqrt{\frac{1}{T_{t_k(q)+1}(q,\pi)}}\right) = 0$ and $T_{t_{k+1}(q)+1}(q,\pi) = T_{t_k(q)+1}(q,\pi)$. Thus overall, we have

$$\sum_{k=0}^{T} \mathbb{E}\left[\mathbb{1}(q \notin \mathcal{L}_{t_{k+1}(q)}, \pi_{t_{k+1}}(q) = \pi) \cdot \min\left(1, \sqrt{\frac{1}{T_{t_k(q)+1}(q,\pi)}}\right) \mid E\right] \leq \sum_{k=1}^{T} \frac{1}{\sqrt{k}} \leq C\sqrt{T}.$$

$\square$

Thus we know that the second difference satisfies

$$\sum_{t=1}^{T} \mathbb{E}[\mathsf{cost}(\mathcal{L}_t, \pi^\star) - \mathsf{cost}(\mathcal{L}^\star, \pi^\star)\mathbb{1}(E^t)] \leq C \cdot \Bigg( B_2 T\delta + LB_2\sqrt{2T\log(8/\delta)}$$

$$+ \frac{L(B_2 - B_1)B_2|\mathcal{Q}|L\log^2(T|\mathcal{Q}|/\delta)}{B_1} \cdot \sqrt{T} \Bigg).$$

Overall by taking $\delta = 1/T$, we know that

$$\mathsf{Regret}(T) \leq \frac{CL(B_2 - B_1)B_2|\mathcal{Q}|L\log^2(T|\mathcal{Q}|)}{B_1} \cdot \sqrt{T}.$$

$\square$

# I  Additional Experiments

## I.1  Synthetic Datasets

We conduct synthetic online and offline experiments for joint optimization of caching and model multiplexing. We use i.i.d. Bernoulli distributions for two models because we want to mimic the model ensemble use case and give a large penalty to the wrong output. In Figure 2, we plot the cumulative cost and regret in online learning for LFU and LEC caching algorithms. We present more data points in Table 3 and Table 4, under the same setting of 10000 requests with 20 different queries and cache size 10. Similar to the real dataset setting, we compare all combinations of caching strategy choices and model multiplexer choices. We consider the frequency distribution as power distribution with $\alpha = 0.5$ and $0.8$. The ground truth cost for each query processed by both models is set as a sample from $r \cdot X + 1$, where $r$ is called as cost ratio and $X$ is a random variable generated from a Bernoulli distribution with the parameter $0.5$. We consider the model multiplexer accuracy with $0.8$ and $1$. We repeat the simulation 1000 times and take the mean. Consistent with Figure 2, our simulation suggests that LEC with a perfect model multiplexer significantly improves the baselines when the cost ratio is large. Simulation 1000 times cannot remove all randomness so we can observe some fluctuations. Theoretically, the columns of choosing model 1 and the columns of choosing model 2 should behave similarly.

| $\alpha$ | cost ratio | selector accuracy | LFU+ model 1 | LFU+ model 2 | LFU+ selector | LEC+ model 1 | LEC+ model 2 | LEC+ selector |
|---|---|---|---|---|---|---|---|---|
| 0.5 | 1.5 | 0.8 | 5.25 | 5.24 | 4.09 | 4.40 | 4.36 | **3.59** |
| 0.8 | 1.5 | 0.8 | 7.61 | 7.60 | 5.93 | 5.73 | 5.68 | **4.85** |
| 0.5 | 1.5 | 1 | 5.25 | 5.24 | 3.31 | 4.40 | 4.36 | **2.74** |
| 0.8 | 1.5 | 1 | 7.61 | 7.60 | 4.81 | 5.73 | 5.68 | **3.68** |
| 0.5 | 100 | 0.8 | 148.05 | 147.38 | 103.43 | 29.93 | **26.83** | 39.32 |
| 0.8 | 100 | 0.8 | 214.93 | 213.88 | 150.31 | 43.77 | **39.02** | 49.88 |
| 0.5 | 100 | 1 | 148.05 | 147.38 | 73.94 | 29.93 | 26.83 | **3.12** |
| 0.8 | 100 | 1 | 214.93 | 213.88 | 107.63 | 43.77 | 39.02 | **4.19** |

Table 3: offline synthetic dataset

| $\alpha$ | cost ratio | LFU+ model 1 | LFU+ model 2 | LFU+ selector | LEC+ model 1 | LEC+ model 2 | LEC+ selector |
|---|---|---|---|---|---|---|---|
| 0.5 | 1.5 | 5.35 | 5.34 | 4.34 | 4.60 | 4.59 | **3.75** |
| 0.8 | 1.5 | 7.79 | 7.78 | 6.32 | 6.01 | 5.98 | **5.09** |
| 0.5 | 100 | 150.93 | 150.37 | 76.80 | 31.88 | 28.65 | **4.85** |
| 0.8 | 100 | 220.19 | 219.49 | 112.26 | 46.44 | 41.45 | **6.31** |

Table 4: online synthetic dataset

## I.2  Real Datasets

In this section, we provide additional experiments on real-world dataset. We run both offline and online algorithms on Lambada dataset and OpenAssistant dataset, with OPT-1.3B vs OPT-13B or FastChat-T5-3B vs Vicuna-13B. We mainly consider three settings:

- In Table 5-12, we consider distinct prompt size 100, total query size 10000, cache size 40, same as the experiments in the main text.

- In Table 13-20, we consider distinct prompt size 1000, total query size 2000, cache size 100.

- In Table 21-28, we consider distinct prompt size 1000, total query size 2000, cache size 0, same as the experiments in the main text.

Our proposed algorithm mostly gives dominant results over other baselines. In Table 21-28, the cache size is set to be 0 so there is no difference between LFU and LEC.

| $\alpha$ | LFU+ large | LFU+ cascade | LFU+ selector | LEC+ large | LEC+ cascade | LEC+ selector |
|---|---|---|---|---|---|---|
| 0.2 | 3.77 | 4.11 | 2.17 | 3.71 | 1.83 | **1.26** |
| 0.5 | 8.11 | 8.85 | 4.54 | 7.93 | 3.51 | **2.29** |
| 0.8 | 11.43 | 12.47 | 6.35 | 10.91 | 4.63 | **2.86** |

Table 5: FLOPs for online lambda dataset, opt-1.3b vs opt-13b

| $\alpha$ | LFU+ large | LFU+ cascade | LFU+ selector | LEC+ large | LEC+ cascade | LEC+ selector |
|---|---|---|---|---|---|---|
| 0.2 | 349.44 | 425.05 | 224.33 | 349.47 | 241.89 | **170.02** |
| 0.5 | 751.78 | 916.58 | 469.74 | 751.77 | 462.34 | **321.46** |
| 0.8 | 1059.95 | 1290.26 | 656.61 | 1060.01 | 596.92 | **397.01** |

Table 6: Latency for online lambda dataset, opt-1.3b vs opt-13b

| $\alpha$ | LFU+ large | LFU+ cascade | LFU+ selector | LEC+ large | LEC+ cascade | LEC+ selector |
|---|---|---|---|---|---|---|
| 0.2 | 1.66 | 1.68 | 0.90 | 1.12 | 0.67 | **0.48** |
| 0.5 | 3.56 | 3.61 | 1.88 | 2.20 | 1.22 | **0.87** |
| 0.8 | 5.01 | 5.09 | 2.64 | 2.75 | 1.49 | **1.04** |

Table 7: FLOPs for online oasst dataset, fastchat-t5 vs vicuna

| $\alpha$ | LFU+ large | LFU+ cascade | LFU+ selector | LEC+ large | LEC+ cascade | LEC+ selector |
|---|---|---|---|---|---|---|
| 0.2 | 9.31 | 13.88 | 7.24 | 8.74 | 8.82 | **5.93** |
| 0.5 | 20.04 | 29.88 | 15.11 | 18.68 | 16.90 | **11.87** |
| 0.8 | 28.24 | 42.12 | 21.14 | 26.07 | 20.31 | **15.49** |

Table 8: Latency for online oasst dataset, fastchat-t5 vs vicuna

| $\alpha$ | selector accuracy | LFU+ large | LFU+ cascade | LFU+ selector | LEC+ large | LEC+ cascade | LEC+ selector |
|---|---|---|---|---|---|---|---|
| 0.2 | 0.8 | 3.49 | 3.81 | 2.60 | 3.44 | **1.50** | 2.00 |
| 0.5 | 0.8 | 7.71 | 8.43 | 5.76 | 7.54 | **3.09** | 3.94 |
| 0.8 | 0.8 | 10.81 | 11.80 | 8.06 | 10.36 | **4.11** | 4.76 |
| 0.2 | 1 | 3.49 | 3.81 | 1.91 | 3.44 | 1.50 | **0.99** |
| 0.5 | 1 | 7.71 | 8.43 | 4.22 | 7.54 | 3.09 | **1.97** |
| 0.8 | 1 | 10.81 | 11.80 | 5.90 | 10.36 | 4.11 | **2.50** |

Table 9: FLOPs for offline lambda dataset, opt-1.3b vs opt-13b

| $\alpha$ | selector accuracy | LFU+ large | LFU+ cascade | LFU+ selector | LEC+ large | LEC+ cascade | LEC+ selector |
|---|---|---|---|---|---|---|---|
| 0.2 | 0.8 | 324.02 | 394.22 | 261.89 | 324.02 | **207.71** | 214.03 |
| 0.5 | 0.8 | 714.87 | 872.60 | 578.91 | 714.87 | **417.01** | 442.05 |
| 0.8 | 0.8 | 1002.26 | 1220.93 | 810.45 | 1002.25 | **538.56** | 549.06 |
| 0.2 | 1 | 324.02 | 394.22 | 197.00 | 324.02 | 207.71 | **141.55** |
| 0.5 | 1 | 714.87 | 872.60 | 435.74 | 714.87 | 417.01 | **287.72** |
| 0.8 | 1 | 1002.26 | 1220.93 | 609.95 | 1002.25 | 538.56 | **358.33** |

Table 10: Latency for offline lambda dataset, opt-1.3b vs opt-13b

| $\alpha$ | selector accuracy | LFU+ large | LFU+ cascade | LFU+ selector | LEC+ large | LEC+ cascade | LEC+ selector |
|---|---|---|---|---|---|---|---|
| 0.2 | 0.8 | 1.54 | 1.55 | 1.09 | 1.01 | **0.56** | 0.66 |
| 0.5 | 0.8 | 3.39 | 3.42 | 2.41 | 2.08 | **1.10** | 1.31 |
| 0.8 | 0.8 | 4.74 | 4.82 | 3.38 | 2.61 | **1.36** | 1.62 |
| 0.2 | 1 | 1.54 | 1.55 | 0.79 | 1.01 | 0.56 | **0.38** |
| 0.5 | 1 | 3.39 | 3.42 | 1.75 | 2.08 | 1.10 | **0.76** |
| 0.8 | 1 | 4.74 | 4.82 | 2.45 | 2.61 | 1.36 | **0.93** |

Table 11: FLOPs for offline oasst dataset, fastchat-t5 vs vicuna

| $\alpha$ | selector accuracy | LFU+ large | LFU+ cascade | LFU+ selector | LEC+ large | LEC+ cascade | LEC+ selector |
|---|---|---|---|---|---|---|---|
| 0.2 | 0.8 | 8.64 | 12.86 | 8.06 | 8.03 | 7.76 | **6.73** |
| 0.5 | 0.8 | 19.07 | 28.42 | 17.81 | 17.66 | 15.63 | **14.14** |
| 0.8 | 0.8 | 26.70 | 39.91 | 24.98 | 24.53 | 19.01 | **18.12** |
| 0.2 | 1 | 8.64 | 12.86 | 6.28 | 8.03 | 7.76 | **4.86** |
| 0.5 | 1 | 19.07 | 28.42 | 13.86 | 17.66 | 15.63 | **10.43** |
| 0.8 | 1 | 26.70 | 39.91 | 19.44 | 24.53 | 19.01 | **13.80** |

Table 12: Latency for offline oasst dataset, fastchat-t5 vs vicuna

| $\alpha$ | LFU+ large | LFU+ cascade | LFU+ selector | LEC+ large | LEC+ cascade | LEC+ selector |
|---|---|---|---|---|---|---|
| 0.2 | 1.81 | 1.97 | 1.62 | 1.80 | 1.72 | **1.52** |
| 0.5 | 3.14 | 3.42 | 2.59 | 3.12 | 3.01 | **2.41** |
| 0.8 | 3.67 | 3.99 | 2.95 | 3.64 | 3.57 | **2.76** |

Table 13: FLOPs for online lambda dataset, opt-1.3b vs opt-13b

| $\alpha$ | LFU+ large | LFU+ cascade | LFU+ selector | LEC+ large | LEC+ cascade | LEC+ selector |
|---|---|---|---|---|---|---|
| 0.2 | 0.17 | 0.21 | 0.17 | 0.17 | 0.19 | **0.17** |
| 0.5 | 0.30 | 0.36 | 0.27 | 0.30 | 0.33 | **0.26** |
| 0.8 | 0.35 | 0.42 | 0.31 | 0.35 | 0.39 | **0.30** |

Table 14: Latency for online lambda dataset, opt-1.3b vs opt-13b

| $\alpha$ | LFU+ large | LFU+ cascade | LFU+ selector | LEC+ large | LEC+ cascade | LEC+ selector |
|---|---|---|---|---|---|---|
| 0.2 | 1.03 | 1.16 | 0.96 | 0.88 | 0.91 | **0.85** |
| 0.5 | 1.78 | 2.01 | 1.52 | 1.45 | 1.47 | **1.27** |
| 0.8 | 2.08 | 2.34 | 1.73 | 1.64 | 1.69 | **1.43** |

Table 15: FLOPs for online oasst dataset, fastchat-t5 vs vicuna

| $\alpha$ | LFU+ large | LFU+ cascade | LFU+ selector | LEC+ large | LEC+ cascade | LEC+ selector |
|---|---|---|---|---|---|---|
| 0.2 | 4.60 | 7.12 | 5.86 | **4.53** | 6.54 | 5.73 |
| 0.5 | 7.98 | 12.33 | 9.33 | **7.86** | 11.27 | 9.08 |
| 0.8 | 9.31 | 14.38 | 10.64 | **9.19** | 13.12 | 10.35 |

Table 16: Latency for online oasst dataset, fastchat-t5 vs vicuna

| $\alpha$ | selector accuracy | LFU+ large | LFU+ cascade | LFU+ selector | LEC+ large | LEC+ cascade | LEC+ selector |
|---|---|---|---|---|---|---|---|
| 0.2 | 0.8 | 1.40 | 1.52 | 1.04 | 1.38 | 1.06 | **0.91** |
| 0.5 | 0.8 | 2.62 | 2.85 | 1.95 | 2.59 | 2.15 | **1.69** |
| 0.8 | 0.8 | 3.09 | 3.37 | 2.30 | 3.05 | 2.62 | **1.98** |
| 0.2 | 1 | 1.40 | 1.52 | 0.76 | 1.38 | 1.06 | **0.57** |
| 0.5 | 1 | 2.62 | 2.85 | 1.43 | 2.59 | 2.15 | **1.13** |
| 0.8 | 1 | 3.09 | 3.37 | 1.68 | 3.05 | 2.62 | **1.35** |

Table 17: FLOPs for offline lambda dataset, opt-1.3b vs opt-13b

| $\alpha$ | selector accuracy | LFU+ large | LFU+ cascade | LFU+ selector | LEC+ large | LEC+ cascade | LEC+ selector |
|---|---|---|---|---|---|---|---|
| 0.2 | 0.8 | 0.13 | 0.16 | 0.11 | 0.13 | 0.12 | **0.10** |
| 0.5 | 0.8 | 0.25 | 0.30 | 0.20 | 0.25 | 0.24 | **0.18** |
| 0.8 | 0.8 | 0.29 | 0.36 | 0.24 | 0.29 | 0.29 | **0.21** |
| 0.2 | 1 | 0.13 | 0.16 | 0.08 | 0.13 | 0.12 | **0.07** |
| 0.5 | 1 | 0.25 | 0.30 | 0.15 | 0.25 | 0.24 | **0.13** |
| 0.8 | 1 | 0.29 | 0.36 | 0.18 | 0.29 | 0.29 | **0.15** |

Table 18: Latency for offline lambda dataset, opt-1.3b vs opt-13b

| $\alpha$ | selector accuracy | LFU+ large | LFU+ cascade | LFU+ selector | LEC+ large | LEC+ cascade | LEC+ selector |
|---|---|---|---|---|---|---|---|
| 0.2 | 0.8 | 0.79 | 0.90 | 0.61 | 0.51 | 0.41 | **0.35** |
| 0.5 | 0.8 | 1.48 | 1.67 | 1.13 | 0.97 | 0.84 | **0.68** |
| 0.8 | 0.8 | 1.75 | 1.97 | 1.34 | 1.13 | 0.99 | **0.79** |
| 0.2 | 1 | 0.79 | 0.90 | 0.45 | 0.51 | 0.41 | **0.23** |
| 0.5 | 1 | 1.48 | 1.67 | 0.84 | 0.97 | 0.84 | **0.46** |
| 0.8 | 1 | 1.75 | 1.97 | 0.99 | 1.13 | 0.99 | **0.54** |

Table 19: FLOPs for offline oasst dataset, fastchat-t5 vs vicuna

| $\alpha$ | selector accuracy | LFU+ large | LFU+ cascade | LFU+ selector | LEC+ large | LEC+ cascade | LEC+ selector |
|---|---|---|---|---|---|---|---|
| 0.2 | 0.8 | 3.55 | 5.49 | 3.42 | 3.41 | 4.43 | **3.15** |
| 0.5 | 0.8 | 6.65 | 10.28 | 6.41 | 6.45 | 8.50 | **5.90** |
| 0.8 | 0.8 | 7.86 | 12.14 | 7.57 | 7.61 | 10.06 | **6.95** |
| 0.2 | 1 | 3.55 | 5.49 | 2.69 | 3.41 | 4.43 | **2.40** |
| 0.5 | 1 | 6.65 | 10.28 | 5.04 | 6.45 | 8.50 | **4.59** |
| 0.8 | 1 | 7.86 | 12.14 | 5.95 | 7.61 | 10.06 | **5.44** |

Table 20: Latency for offline oasst dataset, fastchat-t5 vs vicuna

| $\alpha$ | LFU+ large | LFU+ cascade | LFU+ selector | LEC+ large | LEC+ cascade | LEC+ selector |
|---|---|---|---|---|---|---|
| 0.2 | 4.13 | 4.49 | **2.88** | 4.13 | 4.49 | **2.88** |
| 0.5 | 4.13 | 4.49 | **3.12** | 4.13 | 4.49 | **3.12** |
| 0.8 | 4.13 | 4.49 | **3.21** | 4.13 | 4.49 | **3.21** |

Table 21: FLOPs for online lambda dataset, opt-1.3b vs opt-13b

| $\alpha$ | LFU+ large | LFU+ cascade | LFU+ selector | LEC+ large | LEC+ cascade | LEC+ selector |
|---|---|---|---|---|---|---|
| 0.2 | 0.39 | 0.48 | **0.31** | 0.39 | 0.48 | **0.31** |
| 0.5 | 0.39 | 0.48 | **0.33** | 0.39 | 0.48 | **0.33** |
| 0.8 | 0.39 | 0.48 | **0.34** | 0.39 | 0.48 | **0.34** |

Table 22: Latency for online lambda dataset, opt-1.3b vs opt-13b

| $\alpha$ | LFU+ large | LFU+ cascade | LFU+ selector | LEC+ large | LEC+ cascade | LEC+ selector |
|---|---|---|---|---|---|---|
| 0.2 | 2.50 | 2.66 | **1.72** | 2.50 | 2.66 | **1.72** |
| 0.5 | 2.36 | 2.63 | **1.84** | 2.36 | 2.63 | **1.84** |
| 0.8 | 2.34 | 2.64 | **1.88** | 2.34 | 2.64 | **1.88** |

Table 23: FLOPs for online oasst dataset, fastchat-t5 vs vicuna

| $\alpha$ | LFU+ large | LFU+ cascade | LFU+ selector | LEC+ large | LEC+ cascade | LEC+ selector |
|---|---|---|---|---|---|---|
| 0.2 | 10.45 | 16.08 | **10.37** | 10.45 | 16.08 | **10.37** |
| 0.5 | **10.48** | 16.18 | 11.25 | **10.48** | 16.18 | 11.25 |
| 0.8 | **10.48** | 16.19 | 11.54 | **10.48** | 16.19 | 11.54 |

Table 24: Latency for online oasst dataset, fastchat-t5 vs vicuna

| $\alpha$ | selector accuracy | LFU+ large | LFU+ cascade | LFU+ selector | LEC+ large | LEC+ cascade | LEC+ selector |
|---|---|---|---|---|---|---|---|
| 0.2 | 0.8 | 4.13 | 4.49 | **3.07** | 4.13 | 4.49 | **3.07** |
| 0.5 | 0.8 | 4.13 | 4.49 | **3.07** | 4.13 | 4.49 | **3.07** |
| 0.8 | 0.8 | 4.13 | 4.49 | **3.07** | 4.13 | 4.49 | **3.07** |
| 0.2 | 1 | 4.13 | 4.49 | **2.24** | 4.13 | 4.49 | **2.24** |
| 0.5 | 1 | 4.13 | 4.49 | **2.25** | 4.13 | 4.49 | **2.25** |
| 0.8 | 1 | 4.13 | 4.49 | **2.25** | 4.13 | 4.49 | **2.25** |

Table 25: FLOPs for offline lambda dataset, opt-1.3b vs opt-13b

| $\alpha$ | selector accuracy | LFU+ large | LFU+ cascade | LFU+ selector | LEC+ large | LEC+ cascade | LEC+ selector |
|---|---|---|---|---|---|---|---|
| 0.2 | 0.8 | 0.39 | 0.48 | **0.32** | 0.39 | 0.48 | **0.32** |
| 0.5 | 0.8 | 0.39 | 0.48 | **0.32** | 0.39 | 0.48 | **0.32** |
| 0.8 | 0.8 | 0.39 | 0.48 | **0.32** | 0.39 | 0.48 | **0.32** |
| 0.2 | 1 | 0.39 | 0.48 | **0.24** | 0.39 | 0.48 | **0.24** |
| 0.5 | 1 | 0.39 | 0.48 | **0.24** | 0.39 | 0.48 | **0.24** |
| 0.8 | 1 | 0.39 | 0.48 | **0.24** | 0.39 | 0.48 | **0.24** |

Table 26: Latency for offline lambda dataset, opt-1.3b vs opt-13b

| $\alpha$ | selector accuracy | LFU+ large | LFU+ cascade | LFU+ selector | LEC+ large | LEC+ cascade | LEC+ selector |
|---|---|---|---|---|---|---|---|
| 0.2 | 0.8 | 2.50 | 2.66 | **1.84** | 2.50 | 2.66 | **1.84** |
| 0.5 | 0.8 | 2.36 | 2.63 | **1.79** | 2.36 | 2.63 | **1.79** |
| 0.8 | 0.8 | 2.34 | 2.64 | **1.79** | 2.34 | 2.64 | **1.79** |
| 0.2 | 1 | 2.50 | 2.66 | **1.35** | 2.50 | 2.66 | **1.35** |
| 0.5 | 1 | 2.36 | 2.63 | **1.32** | 2.36 | 2.63 | **1.32** |
| 0.8 | 1 | 2.34 | 2.64 | **1.32** | 2.34 | 2.64 | **1.32** |

Table 27: FLOPs for offline oasst dataset, fastchat-t5 vs vicuna

| $\alpha$ | selector accuracy | LFU+ large | LFU+ cascade | LFU+ selector | LEC+ large | LEC+ cascade | LEC+ selector |
|---|---|---|---|---|---|---|---|
| 0.2 | 0.8 | 10.45 | 16.08 | **10.05** | 10.45 | 16.08 | **10.05** |
| 0.5 | 0.8 | 10.48 | 16.18 | **10.09** | 10.48 | 16.18 | **10.09** |
| 0.8 | 0.8 | 10.48 | 16.19 | **10.10** | 10.48 | 16.19 | **10.10** |
| 0.2 | 1 | 10.45 | 16.08 | **7.91** | 10.45 | 16.08 | **7.91** |
| 0.5 | 1 | 10.48 | 16.18 | **7.94** | 10.48 | 16.18 | **7.94** |
| 0.8 | 1 | 10.48 | 16.19 | **7.94** | 10.48 | 16.19 | **7.94** |

Table 28: Latency for offline oasst dataset, fastchat-t5 vs vicuna

