# OpenReview forum: "Towards Optimal Caching and Model Selection for Large Model Inference"
_NeurIPS.cc/2023/Conference — NeurIPS 2023 poster_

### Official Review · Reviewer_rcfj · 2023-06-17

**Soundness:** 3 good
**Presentation:** 3 good
**Contribution:** 3 good
**Rating:** 7
**Confidence:** 3

**Summary:**

The paper presents an innovative approach to mitigating the challenges posed by large language models (LLMs), namely high resource consumption and latency, by employing a cache to store previous queries and a model selector to choose the most efficient model for query processing. The authors propose an optimal algorithm that jointly optimizes both these approaches in offline and online tabular settings, and demonstrate its effectiveness through a series of simulations and real-world dataset experiments. The paper concludes by suggesting potential future research directions in caching and model selection optimization.

**Strengths:**

1) This paper introduces a novel approach to dealing with the resource-intensive nature of LLMs by jointly optimizing the usage of a cache and a model selector. It creatively merges established concepts of caching with model selection.

2) The theoretical grounding and empirical validation of the proposed algorithm underline the quality of the research. The authors have effectively leveraged the Greedy Dual Size with Frequency (GDSF) and Least Expected Cost (LEC) caching algorithms alongside a model selector to achieve optimal rates.

3) The paper is well-structured and effectively communicates complex ideas and methodologies in a clear and comprehensive manner. The stepwise progression from basic principles to specific techniques is particularly commendable.

4) Addressing the issues of resource consumption and latency in LLM deployment is of critical significance in modern AI. The proposed solutions have a potentially broad impact, improving efficiency and feasibility of real-world AI applications.

**Weaknesses:**

1) Model Selection Scope: The experiments seem to consider a limited number of model options. The approach may encounter difficulties when scaling up to situations where selection is to be made from thousands or millions of models, which is a realistic scenario in complex AI systems.

2) Prompt Representation in Caching: While the paper does a good job of exploring how to insert into a cache, it doesn't delve into the challenge of representing each prompt in the cache. This is an important aspect of LLM serving, as a comprehensive strategy for hashing, differentiating knowledge, and specifying unit information is necessary for efficient caching.

3) The idea and each problem is well known. I am not sure about the novelty of the approach. The application and problem is critical though.

**Questions:**

1)  How does the proposed algorithm scale when selecting from a large number of models, possibly in the order of thousands or millions? Can you provide any theoretical or experimental insight on this scenario?

2) While you discuss inserting prompts into a cache, the paper does not fully address how prompts are represented in the cache. Could you elaborate on this aspect of your approach?

**Limitations:**

The authors have touched upon potential future work related to this study, highlighting open problems that require further investigation. However, a discussion on the limitations of their approach and its potential negative societal impacts seems to be missing. An exploration of the potential risks associated with the approach, such as the possible amplification of biases if the model selector disproportionately selects certain models, would be beneficial. Further, a discussion on how this approach could contribute to increasing centralization and monopolization in AI due to the computational requirements of maintaining and selecting from large model ensembles could offer a balanced perspective.

---

> ### Author Rebuttal · Authors · 2023-08-06
>
>
> Thank you for your valuable comments and suggestions. Please find our responses to each comment below.
>
> ## Comment 1
>
> **Reviewer:**
>
>
> > Model Selection Scope: The experiments seem to consider a limited number of model options. The approach may encounter difficulties when scaling up to situations where selection is to be made from thousands or millions of models, which is a realistic scenario in complex AI systems. How does the proposed algorithm scale when selecting from a large number of models, possibly in the order of thousands or millions? Can you provide any theoretical or experimental insight on this scenario?
>
> **Response:**
>
> Thank you for your comments! We briefly discuss how to generalize to multiple models in Appendix C. If there's $K$ models, we can train a neural network with $K$ dimensional output, each predicting the cost for one of the models. So as the number of models $K$ grows, only the last layer of the neural network scales linearly with $K$. However, this won't be able ideal to deal with the case of millions of the models, since that require a large amount of training data for the selector to estimate accurately the cost for all the models.
>
> Given the current size of large langauge models (billions of parameters -> tens of Gb per model size and memory consumption), it is not yet realistic to simutaneously run thousands or millions models at a time for serving. Also, there may not be that many high-quality language model that needs to be served at the same time. So we believe our setting is a realistic setting especially in the field of large language models.
>
>
>
>
>
>
>
> ## Comment 2
>
> **Reviewer:**
>
>
> > Prompt Representation in Caching: While the paper does a good job of exploring how to insert into a cache, it doesn't delve into the challenge of representing each prompt in the cache. This is an important aspect of LLM serving, as a comprehensive strategy for hashing, differentiating knowledge, and specifying unit information is necessary for efficient caching. While you discuss inserting prompts into a cache, the paper does not fully address how prompts are represented in the cache. Could you elaborate on this aspect of your approach?
>
>
>
> **Response:**
>
> Thank you for the comments! We agree that prompt representation is very important problem left open. There have been some preliminary solutions from vector database in [1], which represents the query as a vector from the embedding of a pre-trained or fine-tuned large language model specifically designed for retrieval. And one can compute the cosine similarity between the vectors of prompts to determine whether they're similar in semantic meaning or not. However, we still don't see a comprehensive research study on which method leads to best caching problem.
>
> For this paper, we focus on the optimal algorithm for caching and model selection. We assume that we either do exact match (with simple hashing on the prompts) or the existing semantic matching algorithms are good. The exact match case can also be applied to practical scenarios, especially when the large language models are used for serving API calls enterprise softwares and the prompts are simple and more repetitive. In our experiments, we focus on exact match case and directly use hashing to represent the prompts for caching. We believe that the appropriate prompt representation deserves a serious and comprehensive study, but this may be out of the scope of our current focus.
>
>
> [1] GPTCache: https://github.com/zilliztech/GPTCache
>
> ## Comment 3
>
> **Reviewer:**
>
>
> > The idea and each problem is well known. I am not sure about the novelty of the approach. The application and problem is critical though.
>
>
> **Response:**
>
> Thank you for your comments!
>
>
> - In terms of formulation, we are the first to formulate the joint optimization of caching and model selection for large language models. Different from most of the traditional caching problems, where the cost of each query is the same, the queries for large language model differ a lot in terms of the response lengths, and thus leading to varying costs. Furthermore, when we have multiple models, this would lead to further variations in the cost for generating responses. Thus the formulation we propose is new.
>
> - Theoretically, we prove the minimax-optimality of the LEC + model selector idea in both offline and online settings. The upper bound analysis is very different from traditional bandit problems. In traditional bandit, one has the ability to choose which action to explore in each round. However, in our setting, the actions (queries) come passively as samples from a fixed query distribution, and we can only choose caches which hurt the exploration of cached queries. This brings a brand new theoretical question and requires completely new analysis.
>
> - We show that the optimal model selector requires a lower confidence bound adjustment on the estimated performance. Such a correction term is new and critical for the optimality of the algorithms in both offline and online settings.
>
> - Empirically, we are the first to systematically benchmark the existing caching and multi-model serving ideas for large language models. We demonstrate the superiority of both LEC and the proposed model selector compared with existing ideas like LFU or cascading.
>
>
>
> ## Comment 4
>
> **Reviewer:**
>
>
> > A discussion on the limitations of their approach and its potential negative societal impacts seems to be missing.
>
>
>
> **Response:**
>
> Thank you for your comments! We will add more discussions on the limitation of the paper in the revised version, including potential biases introduced due to caching and model selector, and how this approach could lead to centralization and monopolization in AI due to the computational requirements of maintaining and selecting from large model ensembles.

---

> > ### Comment · Reviewer_rcfj · 2023-08-18
> >
> > Thank you for the comments. reviewed other comments as well. This is a decent paper. I recommend an accept.

---

### Official Review · Reviewer_X4Ge · 2023-07-06

**Soundness:** 3 good
**Presentation:** 3 good
**Contribution:** 3 good
**Rating:** 8
**Confidence:** 1

**Summary:**

* this paper proposes a theoretically optimal algorithm for efficient large-scale deployment of LLMs
* particularly; they combine caching and model selection to reduce the inference cost
* they provide theoretical guarantees for both caching w and w/o model selection

**Strengths:**

* large scale models are deployed everywhere and the motivation to make inference more efficient is very strong
* the paper comes with strong theoretical guarantees
* the method's efficiency gains are demonstrated in relevant real-world settings using models up to 13B
* the idea to perform model selection on the fly in an online fashion seems clever

**Weaknesses:**

I have to admit that this paper is out of my depth. The considered (problem, solution) seems very convincing and relevant to me, but I'm not familiar with its literature and cannot identify any obvious weaknesses.

**Questions:**

* how can we assess whether caching is relevant in a particular setup? for example, when using publicly available datasets like OpenAssistant; what if there are no redundant queries?

**Limitations:**

yes, the authors discussed limitations and future work directions

---

> ### Author Rebuttal · Authors · 2023-08-04
>
> Thank you for your valuable comments and suggestions. Please find our responses to your comment below.
>
> ## Comment 1
>
> **Reviewer:**
>
> > How can we assess whether caching is relevant in a particular setup? for example, when using publicly available datasets like OpenAssistant; what if there are no redundant queries?
>
>
> **Response:**
>
> Thank you for your comment! The cache hit rate depends heavily on the application. Below are three cases:
>
> - According to [1], the cache hit rate in web search systems is usually 30%-60%.
> - According to our own calculation from the public data in [2], the cache hit rate for saving 3k conversations in real-world chatbot over 33k chat data is 31%. There may not be too many queries that are exactly the same. However, one can match the queries with the same semantic meaning. Thus we use fuzzy matching by constructing a vector database for all the queries, and match them when their cosine similarity is large enough so that they have very close semantic meaning.
> - When it comes to using LLM for API calls to software in enterprise, we shall expect more redundant queries and the cache hit rate might be much higher than chat even if we do exact match.
>
>
> [1] Jeff Dean, Building Software Systems at Google and Lessons Learned
>
> [2] Lianmin Zheng et al., Judging LLM-as-a-judge with MT-Bench and Chatbot Arena

---

> > ### Comment · Reviewer_X4Ge · 2023-08-16
> > **Thank you.**
> >
> > Thanks for your responses. Would be great if you could incorporate this cache hit discussion in the final version. I recommend acceptance.

---

### Official Review · Reviewer_FCsT · 2023-07-07

**Soundness:** 3 good
**Presentation:** 2 fair
**Contribution:** 2 fair
**Rating:** 5
**Confidence:** 3

**Summary:**

This paper presents an LLM inference framework design that aims to reduce inference cost by caching and model selection. This work first provides an optimal formulation for jointly optimizing both caching and model selection in both offline and online settings. Then, evaluations on simulations and two tasks show that proposed work achieves cost saving compared to baseline.

**Strengths:**

(1) The high-level idea of caching + model selection sounds reasonable for LLM inference.

(2) Detailed formulation of the optimization problem.


**Weaknesses:**

(1) My most unsatisfying part of this work is that the problem that authors try to solve should not be simplified to “cost saving” only, but it should be formulated as “a cost-accuracy trade-off”. I do see that the authors acknowledge that it is a trade-off (e.g., “The need for fuzzy search.” In introduction, and in section 5.2 “If the small model is chosen but its result is wrong, the large model must be run and it will incur an additional penalty.”), but I really want the authors to investigate more about this instead of simply comparing on the cost. For example, for fuzzy search during caching, how much will it actually affect inference accuracy? For model selection, what if there are more than two models, and what if there is no way to know whether the inference result is correct or not (so that you cannot use multiple models). Ultimately, the results should be presented as a cost-accuracy Pareto curve, and the proposed work has to show how it advances the Pareto frontier compared to baseline (achieving better accuracy under same cost, or achieving same accuracy under less cost). The unbalance between formulation (5 pages) and experiments (1.5 pages) exacerbates this issue.

(2) In introduction it says “For the fuzzy search problem, semantic search or vector-embedding-based ideas provide a systematic solution that includes embedding extraction and matching algorithms. To simplify the problem, we assume that there exists some semantic search oracle that can group the prompts with the same semantic meaning…”. To me this assumption oversimplify the caching problem. In order to demonstrate the feasibility of caching, it’s needed to actually build yourself a semantic search-based cache, and actually evaluate how it affects the accuracy-cost tradeoff.

(3) In section 5.2 it says “We fine-tune a BERT-base model as the model selector by predicting whether the small model can give the correct result and achieve 80.2% accuracy.”. But I can’t find any further information such as the fine-tune hyperparameters, and whether this 80.2% accuracy is training accuracy or validation accuracy.

**Questions:**

(1) In section 5.2, the next-token prediction task uses FLOPs as the cost, and the chat assistant task uses latency as the cost. Is there a reason to use different cost definition here? Or why not evaluate both cost definitions in both cases?

**Limitations:**

The authors acknowledged some limitations, such as “We leave the training of a better selector as future work.”.

---

> ### Author Rebuttal · Authors · 2023-08-06
>
> Thank you for your valuable comments and suggestions. Please find our responses to each comment below.
>
> ## Comment 1
>
> **Reviewer:**
>
>
> > That authors try to solve should not be simplified to “cost saving” only, but it should be formulated as “a cost-accuracy trade-off”. I really want the authors to investigate more about this instead of simply comparing on the cost. The results should be presented as a cost-accuracy Pareto curve, and the proposed work has to show how it advances the Pareto frontier compared to baseline (achieving better accuracy under same cost, or achieving same accuracy under less cost).
>
> **Response:**
>
> It is a misunderstanding that we only focus on "cost saving". In both theory and experiments, we focus on the **cost-accuracy trade-off rather than cost**. Additionally, our framework is not limited to cost-accuracy trade-off. It optimizes for the objective, which can be defined as any reward depending on the scenario. In small-large model selection which is the motivating example in our paper, the reward is defined as the cost-accuracy trade-off. While in the model ensemble setting, the objective can be simply defined as accuracy. See below for more details.
>
> - It might be misleading that we call $c$ as a cost function (a terminology in bandit), which is indeed a trade-off term between the cost in real world and accuracy. For example, a common choice of $c$ can be chosen as $c(q) =$ cost of $q - \lambda\times$ accuracy of $q$. Our theoretical analysis shows that once we choose the target function $c$ regarding how one combines cost and accuracy (i.e. fix a $\lambda$), the proposed algorithms always give the minimax-optimal trade-off between cost and accuracy. Thus our framework is general enough to incorporate any trade-off between cost and accuracy.
>
> - In our experiments, we **always guarantee the output quality at the same level**, and compare the cost. Thus the proposed algorithm advances the Pareto frontier compared to the baseline. In all experiments, if we call small model and find that it is not giving satisfying result (the evaluation score from GPT4 is smaller than 6 out of 10), we will call the large model again to fix the output. This incurs extra cost due to the error induced and the extra compute needed to call the large model. No matter whether we hit or miss the cache, and whether we select the wrong model or the correct model, the output will always be satisfying with score larger than 6. One may also change the reward from binary satisfying vs unsatisfying to the actual scalar of GPT 4 eval.
>
>
> ## Comment 2
>
> **Reviewer:**
>
> >  For model selection, what if there are more than two models, and what if there is no way to know whether the inference result is correct.
>
> **Response:**
>
> We briefly discuss how to generalize to multiple models in Appendix C. If there're $K$ models, we can train a neural network with $K$  dimensional output, each predicting the cost for one model. If there is no way to evaluate the result, the model selection algorithm would not work. However, there have been a good amount of evaluation methods, like judgement from GPT4, or a reward model trained from human preference data.
>
> ## Comment 3
>
> **Reviewer:**
>
> > In introduction “We assume that there exists some semantic search oracle that can group the prompts with the same semantic meaning…”. This assumption oversimplify the caching problem. In order to demonstrate the feasibility of caching, it’s needed to build a semantic search-based cache, and evaluate how it affects the accuracy-cost tradeoff. For fuzzy search during caching, how much will it actually affect inference accuracy?
>
> **Response:**
>
> - There have been some preliminary studies and systems using vector database for caching in [1], which represents the query as a vector from the embedding of a pre-trained or fine-tuned large language model specifically designed for retrieval. Thus the effectiveness of simple caching has been demonstrated, and building such a semantic search system only for caching is not our main focus given existing efforts.
>
> - According to [2], the cache hit rate in web search systems is usually 30%-60%. The semantic matching part in large models shall be very similar to the existing ones used in the web search systems. And thus there are been mature systems in industry that validate the effectiveness of caching system.
>
> - Even without semantic search and fuzzy matching, the exact-match-based caching is useful in practice. When it comes to using LLM for API calls to software, we expect more redundant and identical queries. And the cache hit rate can be much higher than chat even if we do exact match. Our experiments with exact match also validates the effectiveness of the proposed algorithms.
>
> - In our paper, we focus on identifying the information-theoretic optimal algorithms for jointly optimizing the caching and model selection system. We believe that the most appropriate fuzzy search deserves a serious and comprehensive study. But this may be out of the scope of our current focus.
>
> [1] GPTCache: https://github.com/zilliztech/GPTCache
>
> [2] Jeff Dean, Building Software Systems at Google and Lessons Learned
>
> ## Comment 4
>
> **Reviewer:**
>
> > - In section 5.2 it says “We fine-tune a BERT-base model and achieve 80.2% accuracy.”. But I can’t find any further information such as the fine-tune hyperparameters, and whether this 80.2% accuracy is training accuracy or validation accuracy.
> > - In section 5.2, the next-token prediction task uses FLOPs as the cost, and the chat assistant task uses latency as the cost. Is there a reason to use different cost definition here? Or why not evaluate both cost definitions in both cases?
>
> **Response:**
>
> - We will include all the details on the fine-tune hyperparameters and the models. This 80.2% accuracy is test accuracy on the unseen prompts.
>
> - We have included experiments for both FLOPs and latency for both cases in the revised paper, along with the one page PDF.

---

> > ### Comment · Reviewer_FCsT · 2023-08-15
> > **Rebuttal Reply**
> >
> > I appreciate the clarifications and additional experiment results from the author. They do help make it easier to me to appreciate this work, thus I'm raising my score to borderline accept. I would highly recommend the authors to always keep in mind of balancing the theory/formulation and engineering/experiments in future paper writing, not only for improving the soundness of the paper itself, but also for improving the chance of being actually used by AI community, a community that currently emphasizes engineering efforts and practical usability.

---

> > > ### Author Response · Authors · 2023-08-15
> > >
> > > Thank you for your great suggestions! We are working on building real product that can benefit the AI community more based on the paper. We will release it soon once finished. Please stay tuned!

---

### Official Review · Reviewer_puCe · 2023-07-08

**Soundness:** 4 excellent
**Presentation:** 4 excellent
**Contribution:** 3 good
**Rating:** 7
**Confidence:** 3

**Summary:**

The paper addresses the computation costs of large language models. Specifically, the paper proposes a framework where previous queries are cached and retrieved and where given a query to process, an appropriate model is selected to answer based on the query. The framework flow is the following: given a query in test time, we check whether the response can be retrieved from the cache. If so, we return from the cache. If not, we select one of two models based on the query, where a natural configuration is to have two models - a large more accurate model with large computation cost and a small less accurate model with smaller computation costs (the com. cost can be measured in various ways).


Cache:
The method estimates two oracle - the first oracle DenEstOracle estimates the probability distribution that a query q will be observed. This is useful in defining the cache, sense we want to cache the most frequent queries.  The second oracle is RegressionOracle which estimates the cost of processing a query. This is useful since we want to cache the queries which cost the most to process. Then, in the online setting, in time step q we can have a cache which in stores the most frequent queries which also have the most cost. Remember that the cache is finite, so formally, in time step t the cache which is denoted L_t holds the L caches such that for query q in L_t the value P_t(q) * Cl_t(q) is larger than all previously seen queries that are not in the cache. This basically means that the set of queries in the cache is optimal, in the sense that the estimated cost of encountering those queries is the largest given both the probability that we will encounter them and the cost of processing them.

Model selector:
Now, for the model selector, the method assume that the cost of a query is weighed sum of the C_0(q) + Y(q) * C_1(q), where C_0(q) is the compute cost of the small model assuming the user is satisfied with the results, C_1(q) is the cost if the user is not satisfied with the results and Y(q) is a random binary variable denoting whether or not the user is satisfied. Appendix A discusses the costs and model selection and considers options such as C_1(q) == cost of user dissatisfaction or C_1(q) == cost of re-running the large model on q where the latter is used in the online experiment presented in the paper.

Now, in the online setting the model selector simply chooses the model with the smaller cost and the cache stores the queries with largest P(q) * min(C*_s(q), C*_l(q)) where C* is the estimated cost for the RegressionOracle.

The paper provides experimental results in synthetic, offline and online setting (online setting is done using the open assist dataset). The paper compares against baseline such as simple Least Frequently Used (LFU) cache as baseline cache and simple cascade (calling the large model if the result of the small model is un-satisfactory) as model selector. The paper compares the baselines against the proposed cache and model selector, separately and jointly.

**Strengths:**

* The paper addresses an important real-world problem in a novel manner with both a detailed theoretical discussion and practical experiments and results.
* The presented framework is general and can be extended/adjusted to various use cases and practical applications.
* The paper is well written and easy to follow.

**Weaknesses:**

IMO, the experiments can benefit from further analysis, e.g.:
* Performance/accuracy analysis as function of number of steps in the online setting.
* What is the cost of running the model selector and how does that affect the performance of the system?
* Can you provide some accuracy measure of the oracles? for example, in the online setting, how many steps are required so that the oracles would be properly estimated? What is the accuracy as a function of the step number?


**Questions:**

Can you discuss how to ensure the cache diversity? the top most probable queries might be very similar to each other. I understand that the method does not cacher queries, but "query groups" where similar queries are cached as one group (it is assumed in the paper that there exists some semantic search oracle that can group the prompts with the same semantic meaning). Still, due to the long tail distribution of queries, the bottom most frequent queries in the cache might be less similar to each other than the top ones, which might motivates grouping the queries differently, i.e. taking into account P(q) and varying the minimal distance to group two queries together by semantic search oracle that groups prompts close embedding. If the cache is not diversified enough, less common queries would be processed slower. This is especially true if we want to ensure the system is not biased towards specific user population.


**Limitations:**

The authors did not specifically addressed limitations or potential negative societal impact of their work.

---

> ### Author Rebuttal · Authors · 2023-08-06
>
> Thank you for your valuable comments and suggestions. We have corrected all the typos and added all the suggested details in the revision. Please find our responses to each comment below.
>
>
>
> ## Comment 1
>
> **Reviewer:**
>
>
> > The experiments can benefit from further analysis, e.g.:
> - Performance/accuracy analysis as function of number of steps in the online setting.
> - What is the cost of running the model selector and how does that affect the performance of the system?
> - Can you provide some accuracy measure of the oracles? for example, in the online setting, how many steps are required so that the oracles would be properly estimated? What is the accuracy as a function of the step number?
>
>
> **Response:**
>
> Thank you for your comments!
>
> - For the online case, the accuracy w.r.t. step size largely depends on the number of distinct queries we face. In our experiments, we use 100 distinct queries. And thus the oracle quickly converges after seeing the cost most of the queries once. The accuracy is thus proportional to the fraction of seen queries, and is directly affected by the distribution of the queries. We are happy to add more details in the revised paper.
>
> - The cost of running the model selector is equivalent to one forward call of the used model (BERT or causal language models). In contrast, for a response with 1000 tokens, the causal language models like LLama, Vicuna and GPT 3.5 require 1000 times of forward call. Thus the cost of running the model selector is neglibile compared to the cost used to generate the responses.
>
> - The shaded area in Figure 2 shows the variance which shows the convergence rate of the oracle selector. In practice, the quality of the predictor continues to improve as we have more online data. And we can have good estimation for those that are seen multiple times (and more likely saved in cache).
>
>
>
>
> ## Comment 2
>
> **Reviewer:**
>
>
> > Can you discuss how to ensure the cache diversity? the top most probable queries might be very similar to each other. I understand that the method does not cache queries, but "query groups" where similar queries are cached as one group (it is assumed in the paper that there exists some semantic search oracle that can group the prompts with the same semantic meaning). Still, due to the long tail distribution of queries, the bottom most frequent queries in the cache might be less similar to each other than the top ones, which might motivates grouping the queries differently, i.e. taking into account P(q) and varying the minimal distance to group two queries together by semantic search oracle that groups prompts close embedding. If the cache is not diversified enough, less common queries would be processed slower. This is especially true if we want to ensure the system is not biased towards specific user population.
>
>
>
> **Response:**
>
>
> Thank you for the comment! We are happy to include more discussions on the cache diversity. You are absolutely correct that in the case when the bottom most frequent queries in the cache are less similar to each other, we may want to motivate grouping differently. Vector database is popular for retrieving relevant documents by matching queries with similar embeddings into the same group, thus grouping responses with similar semantic meaning. In the case of caching, we may want to design new embedding method that takes into account the frequency $P(q)$ and design different thresholds for grouping. This will be a very interesting open problem to be further explored.
>
>
>
>
> ## Comment 3
>
> **Reviewer:**
>
>
> > The authors did not specifically addressed limitations or potential negative societal impact of their work.
>
>
>
> **Response:**
>
> Thank you for your comments! We will include more discussions on the limitations or potential negative societal impacts of the work. For example, this may lead to biased processing speed towards specific user population when the group sends more queries than other groups.

---

> > ### Comment · Reviewer_puCe · 2023-08-20
> >
> > I would like to thank the authors for the clarifications. I keep my original score of 7, this is a good paper IMO and would benefit the community and I would recommend accepting it.

---

### Official Review · Reviewer_13j8 · 2023-07-22

**Soundness:** 3 good
**Presentation:** 3 good
**Contribution:** 3 good
**Rating:** 6
**Confidence:** 3

**Summary:**

The paper investigates the combination of caching and model selection strategies to reduce the inference costs of large models (LLMs in particular). In the result, the authors propose a theoretically grounded algorithm that demonstrates promising results in practice.

**Strengths:**

1. The problem of the cost-effective LLM inference is in high demand.
2. The paper is well written and easy to follow.
3. Theoretical contribution seems novel and complete.
4. Synthetic experiments are promising, especially when the ratio between min/max query costs is large.

**Weaknesses:**

In my opinion, Section 5.2 lacks some important details and clarifications. It makes it difficult to estimate the practical contribution of the proposed approach. I highly recommend adding the detailed evaluation protocols for both tasks to the appendix.

**Minor**
* In Figure 2, please specify which plot corresponds to the offline/online setting.
* In L132 and L152, one can introduce $\mathcal{L}^{\star}$ and $\pi^{\star}$ that appear in the corresponding equations.
* Consider adding the cost measures to the captions in Tables 1,2
* Consider specifying the hardware specs used for the latency evaluation and corresponding std values in Table 2.
* Typos: L168 "last year" -> "last layer"

**Questions:**

The following questions will help to address my concerns about the empirical contribution in Section 5.2:

* How is the performance measured for each task?
* For a fair comparison, all methods within each row (Tables 1,2) should provide the same model quality.
    * Does this hold true in both cases? If yes, how is it guaranteed for the chat assistant task?
    * What is the target performance for each task?
    * What is the criterion to quantify if the output of the small model satisfies or not (especially in the chat assistant task)?
* In the offline setting, the predictor is not accurate enough to outperform "cascade" while it is not the case in the online setting. I am curious why it happens.
    * Is the chat assistant task easier for the selector? Or is it caused by the difference in the online/offline settings?
    * What selector is used in the online setting? How accurate is it at convergence? What portion of queries does the predictor observe until convergence?
   * What if one tries the online and offline settings for the Lambada and chat assistant tasks, respectively?
* How do the costs and gains depend on the cache size? What are the cache sizes in both settings? How are they selected?
* It might be useful to report the "large", "cascade," and "selector" costs without caching to understand the LFU/LEC gains better.

**Limitations:**

Limitations are well discussed.

---

> ### Author Rebuttal · Authors · 2023-08-05
>
> Thank you for your valuable comments and suggestions. We have corrected all the typos and added all the suggested details in the revision. Please find our responses to each comment below.
>
> ## Comment 1
>
> **Reviewer:**
>
>
> > In my opinion, Section 5.2 lacks some important details and clarifications. It makes it difficult to estimate the practical contribution of the proposed approach. I highly recommend adding the detailed evaluation protocols for both tasks to the appendix.
>
> **Response:**
>
> Thank you for your comments and sorry for missing the details! We will include all the detailed evaluation protocols in the Appendix. Please find below for our responses to the individual questions.
>
>
>
> ## Comment 2
>
> **Reviewer:**
>
>
> > How is the performance measured for each task? What is the target performance for each task? What is the criterion to quantify if the output of the small model satisfies or not (especially in the chat assistant task)?
>
>
>
> **Response:**
>
> For the offline next-token prediction task, the target performance metric is the number of correct tokens predicted. We have the ground-truth token from the Lambada dataset. Thus it is easy to measure the success. If the small model is chosen but its result is wrong, the large
> model must be run and it will incur an additional FLOPs.
>
> For the online chat assistant task, the quality of response is evaluated by GPT4 judgement. We say a response is satisfying if the score is larger than 6 out of 10, and unsatisfying otherwise. If the response from the small model is unsatisfying, we will call the large model again and incur an additional cost in latency.
>
> ## Comment 3
>
> **Reviewer:**
>
>
> > For a fair comparison, all methods within each row (Tables 1,2) should provide the same model quality.
> Does this hold true in both cases? If yes, how is it guaranteed for the chat assistant task?
>
> **Response:**
>
> Thank you for your comments! Yes, in the next-token prediction task, if the small model fails to predict the ground-truth, we will call the large model again to re-generate. Thus it provides the same quality. In the chat assistant task, if the response of the small model is evaluated as unsatisfying (score less than 6 out of 10 in GPT 4 judgement), the large model will be called to ensure the output quality is as good, at a cost of higher latency due to calling both small and large model sequentially. Thus all the methods are providing the same model quality. One may adjust the GPT 4 judgement score to be more strict on the quality of the output.
>
>
>
>
> ## Comment 4
>
> **Reviewer:**
>
>
> > In the offline setting, the predictor is not accurate enough to outperform "cascade" while it is not the case in the online setting. I am curious why it happens. Is the chat assistant task easier for the selector? Or is it caused by the difference in the online/offline settings? What selector is used in the online setting? How accurate is it at convergence? What portion of queries does the predictor observe until convergence? What if one tries the online and offline settings for the Lambada and chat assistant tasks, respectively?
>
>
> **Response:**
>
> Thank you for your comments! In both offline and online settings, we work with 100 distinct prompts. In offline setting, we train a BERT-based model for predicting based on 2k prompt-responses pairs. In the online setting, we consider a tabular case selector where one memorizes the cost for each prompt after seeing them once, and thus the selector converges when it sees each of the query once.
> In the online setting, the initialization for the selector is calling the small model first (same with cascade) and changing to call the larger model when it learns that the larger model is better. This makes our method better than cascade.
>
> In practice, the quality of the predictor continues to improve as we have more online data. And we can have good  estimation for those that are seen multiple times (and more likely saved in cache). The shaded area in Figure 2 shows the variance which shows the convergence rate.
>
> We added new experiments of the online and offline settings for the Lambada and chat assistant tasks which are included in the attached pdf.
>
>
>
>
> ## Comment 5
>
> **Reviewer:**
>
>
> > How do the costs and gains depend on the cache size? What are the cache sizes in both settings? How are they selected?
> It might be useful to report the "large", "cascade," and "selector" costs without caching to understand the LFU/LEC gains better.
>
>
> **Response:**
>
> Thank you for your comments! The gains for caching largely depend on the number of cached items and the distributions of real-world queries. In our experiments, we work with 100 distinct queries and cache 40 queries. Thus the gain for caching is relatively large compared to the case without caching. If we only save 5 queries, the gain for caching can be lower.
>
> To provide a more comprehensive comparison, we have added new experiments for different cache size, including no cache, 40 cache out of 100 queries, and 100 cache out of 1000 queries in the revised version (along with the uploaded PDF).

---

> > ### Comment · Reviewer_13j8 · 2023-08-10
> > **Response to the rebuttal**
> >
> > I would like to thank the authors for their thoughtful clarifications and additional results. The setting with 1k distinct queries out of 2k and 100 cache size sounds interesting and reasonable, so I look forward to seeing these results in the revision.
> >
> > Overall, my questions have been well addressed. So, I'm happy to update my score accordingly.

---

### Author Rebuttal · Authors · 2023-08-08

We would like to thank the reviewers for the valuable comments and suggestions, which have helped us greatly improve the paper. Here we briefly summarize our additional experiments done for the rebuttal period.

In our original experiments in the main text, we include 2 experiment tables with $100$ distinct queries, $40$ cache size and $10k$ total number of queries on both FLOPs for offline next-token-prediction task and latency for online chat task.

In our revised version, we include **22 extra experiment tables on different parameters and tasks**, listed as below:

- For the original set of parameters ($100$ distinct queries, $40$ cache size and $10k$ total number of queries), we include 6 extra experiment tables. Combined with the two original tables, they form the metric [FLOPs, latency] on [offline, online]  [next-token-prediction, chat] task, thus in total 8 tables.

- We include another 8 tables with parameters ($1000$ distinct queries, $0$ cache size and $2000$ total number of queries), with the metric [FLOPs, latency] on [offline, online]  [next-token-prediction, chat] task.

- We include another 8 tables with parameters ($1000$ distinct queries, $100$ cache size and $2000$ total number of queries), with the metric [FLOPs, latency] on [offline, online]  [next-token-prediction, chat] task.

Due to space limit of one page PDF for uploading during rebuttal, we only include the first 6 extra experiments on the same parameters for different metrics and different tasks / settings. We also include two extra tables on the case without any cache ($1000$ distinct queries, $0$ cache size and $2000$ total number of queries) below. The rest will be available in the revised paper. The tables list cumulative costs ($10^3$).

> FLOPs on offline lambda dataset, opt-1.3b vs opt-13b

| \(\alpha\) | selector accuracy | large | cascade | selector |
|------------|-------------------|------------|--------------|---------------|
| 0.2        | 0.8               | 4.07       | 5.65         | **3.22**     |
| 0.5        | 0.8               | 4.16       | 4.57         | **3.02**     |
| 0.8        | 0.8               | 4.12       | 4.42         | **3.09**     |
| 0.2        | 1                 | 4.14       | 3.82         | **1.87**     |
| 0.5        | 1                 | 4.13       | 4.35         | **2.28**        |
| 0.8        | 1                 | 4.13       | 4.46         | **2.24**     |


> Latency on offline lambda set, opt-1.3b vs opt-13b

| \(\alpha\) | selector accuracy | large | cascade | selector |
|------------|-------------------|------------|--------------|---------------|
| 0.2        | 0.8               | 0.39       | 0.54         | **0.35**     |
| 0.5        | 0.8               | 0.39       | 0.48         | **0.32**         |
| 0.8        | 0.8               | 0.39       | 0.48         | **0.32**         |
| 0.2        | 1                 | 0.39       | 0.47         | **0.19**     |
| 0.5        | 1                 | 0.39       | 0.45         | **0.23**     |
| 0.8        | 1                 | 0.39       | 0.46         |**0.24**         |

---

### Decision · Program_Chairs · 2023-09-21

**Decision:**

Accept (poster)

**Comment:**

This paper presents an algorithm for speeding up inference of large models via two methods: 1) caching prior queries, and 2) selecting an appropriate model that minimizes cost while attaining reasonable performance. An algorithm is developed to perform caching and model selection optimally, and experiments show that the algorithm provides significant speedups. Reviewers generally agreed about the soundness of the method, and the post-rebuttal consensus was to accept this paper (particularly given the large amount of additional experiments included).